# University of Warsaw Lagrangian Cloud Model (UWLCM) 1.0: a modern Large-Eddy Simulation tool for warm cloud modeling with Lagrangian microphysics

Piotr Dziekan, Maciej Waruszewski, and Hanna Pawlowska

Institute of Geophysics, Faculty of Physics, University of Warsaw, Poland

**Correspondence:** Piotr Dziekan (pdziekan@fuw.edu.pl)

**Abstract.** A new anelastic large-eddy simulation model with an Eulerian dynamical core and Lagrangian particle-based microphysics is presented. The dynamical core uses the MPDATA advection scheme and the generalized conjugate residual pressure solver, while the microphysics scheme is based on the Super-Droplet Method. Algorithms for coupling of Lagrangian microphysics with Eulerian dynamics are presented, including spatial and temporal discretisations and a condensation substepping algorithm. The model is free of numerical diffusion in the droplet size spectrum. Activation of droplets is modeled explicitly, making the model less sensitive to local supersaturation maxima than models in which activation is parameterized. Simulations of a drizzling marine stratocumulus give results in agreement with other LES models. It is shown that in the Super-Droplet Method a relatively low number of computational particles is sufficient to obtain correct averaged properties of a cloud, but condensation and collision-coalescence have to be modeled with a time step of the order of 0.1 s. Such short time steps are achieved by substepping, as the model time step is typically around 1 s. Simulations with and without explicit subgrid-scale turbulence model are compared. Effects of modeling subgrid-scale motion of super-droplets are investigated. The model achieves high computational performance by using GPU accelerators.

## 1   Introduction

In the last decade, Lagrangian particle-based cloud microphysics schemes have been drawing increasing attention. They are similar to Eulerian bin schemes in that they explicitly model the size spectrum of droplets and explicitly resolve microphysical processes, but have a number of advantages over them (Grabowski et al., 2018b). One of the advantages is that Lagrangian schemes have no numerical diffusion in the spectrum of droplet sizes. Several Lagrangian schemes for warm cloud microphysics have been developed so far (Andrejczuk et al., 2008; Shima et al., 2009; Riechelmann et al., 2012). Arguably, the most important difference between these schemes is in the way collision-coalescence is modeled. The coalescence algorithm used in the Super-Droplet Method (SDM) of Shima et al. (2009) seems to be the most promising, as it was found to be the most accurate of the coalescence algorithms used in various Lagrangian microphysics schemes (Unterstrasser et al., 2017, where it is called the "all-or-nothing" algorithm). A numerical implementation of the SDM is a major part of the libcloudph++ library (Arabas et al., 2015) developed by the cloud modeling group at the University of Warsaw.

In this paper, we document development of a new Large-Eddy Simulation (LES) model called the University of Warsaw Lagrangian Cloud Model (UWLCM). It is an anelastic model with a finite-difference Eulerian dynamical core and a Lagrangian microphysics scheme. The Lipps-Hemler anelastic approximation (Lipps and Hemler, 1982) is used, which is applicable to a wide range of atmospheric flows (Klein et al., 2010; Smolarkiewicz, 2011). The dynamical core is implemented using the libmpdata++ software library (Jaruga et al., 2015) also developed by the cloud modeling group at the University of Warsaw. libmpdata++ is a collection of solvers for the generalized transport equation. In libmpdata++, advection is modeled using the multidimensional positive-definite advection transport algorithm (MPDATA) – see Smolarkiewicz (2006) for a recent review. Liquid water is modeled with the SDM implemented in libcloudph++. We do not assume any artificial categorization of liquid water particles. In consequence, all particles, i.e. humidified aerosols, cloud droplets and rain drops, evolve according to the same set of basic equations.

One of the key reasons for developing a new model is to use a modern software development approach. The model code is written in the C++ programming language and makes use of many mature libraries available in that language (e.g. Blitz++, Boost, Thrust). The code is open-source and under a version-control system. A set of automated tests greatly helps in ensuring the correctness of the model. The automated tests include a 2D moist thermal simulation, a 2D kinematic stratocumulus simulation and a test of different combinations of model options. Moreover, modeling of physical processes, e.g. condensation, advection, coalescence, sedimentation, is tested separately by the libmpdata++ and libcloudph++ test suites. UWLCM makes efficient use of modern computers that have both central processing units (CPUs) and graphics processing units (GPUs). The Eulerian computations of the dynamical core are done on CPUs and, simultaneously, Lagrangian microphysical computations are done on GPUs. However, it is also possible to run the Lagrangian microphysics on CPUs.

Some results obtained using earlier versions of UWLCM have already been published. In Grabowski et al. (2018a), UWLCM was used to model a 2-dimensional moist thermal and in Grabowski et al. (2018b), an idealized 3-dimensional cumulus cloud was modeled. Here, we present simulations of a drizzling marine stratocumulus using the DYCOMS-II RF02 setup. UWLCM results are compared with 11 LES models that took part in the Ackerman et al. (2009) intercomparison. Sensitivity of UWLCM to the parameters of the microphysics scheme and to the description of the subgrid-scale (SGS) turbulence is studied. It is of particular interest how much drizzle a LES model with Lagrangian microphysics produces, compared to models with bin or bulk microphysics that took part in the intercomparison. To our knowledge, LES simulations with warm cloud Lagrangian microphysics were used to study drizzling stratocumulus only by Andrejczuk et al. (2008, 2010). This type of models was more often employed to study cumulus clouds (Riechelmann et al., 2012; Naumann and Seifert, 2015; Arabas and Shima, 2013; Hoffmann et al., 2015, 2017).

Section 2 presents the governing equations of the model, section 3 describes the numerical algorithms for solving these equations, the stratocumulus simulation results are discussed in section 4, and section 5 contains a summary and planned developments of the model. A list of symbols used and their definitions are given in appendix A, appendix B compares two substepping algorithms for condensation and appendix C contains a brief description of the software implementation of the model.

## 2 Governing equations

### 2.1 Eulerian variables

Eulerian prognostic variables of the model are the potential temperature $\theta$, the water vapor mixing ratio $q_v$ and the air velocity $\boldsymbol{u}$. Equations governing time evolution of these variables are obtained through the Lipps-Hemler approximation, which relies on the assumption that the atmosphere does not depart far from some stationary state, called the *reference* state (Lipps and Hemler, 1982). The reference state is assumed here to be a dry, hydrostatically balanced state with constant stability $S^r$. $S^r$ is equal to the average stability of the sounding used to initialize a simulation. Surface density and pressure of the reference state are equal to those of the initial sounding. Vertical profiles of potential temperature and density of dry air in the reference state are (Clark and Farley, 1984):

$$\theta^r(z) = \theta_v^0 \exp(S^r z), \tag{1}$$

$$\rho_d^r(z) = \rho^0 \exp(-S^r z) \left[ 1 - \frac{g}{c_{pd} S^r \theta_v^0} \left(1 - \exp(-S^r z)\right) \right]^{(c_{pd}/R_d)-1}, \tag{2}$$

where $\theta_v^0$ and $\rho^0$ are values of the virtual potential temperature and of the air density taken from the initial sounding at the ground level. An auxiliary *environmental* state is introduced to increase accuracy of numerical calculations (Smolarkiewicz et al., 2014, 2019). It is a hydrostatically balanced moist state with stationary profiles $\theta^e(z)$, $p^e(z)$, $T^e(z)$, $q_v^e(z)$ and $q_l^e(z)$ calculated from the initial sounding. If the initial sounding is supersaturated, all supersaturation is assumed to be condensed in the environmental state.

The set of anelastic Lipps-Hemler equations (Lipps and Hemler, 1982; Grabowski and Smolarkiewicz, 1996; Clark and Farley, 1984) that govern time evolution of the Eulerian prognostic variables is

$$D_t \boldsymbol{u} = -\nabla \pi + \boldsymbol{k} B + \boldsymbol{F}_u + \boldsymbol{\mathcal{D}}_u, \tag{3}$$

$$D_t \theta = \frac{\theta^e}{T^e} \left( \frac{l_v}{c_{pd}} C \right) + F_\theta + \mathcal{D}_\theta, \tag{4}$$

$$D_t q_v = -C + F_{q_v} + \mathcal{D}_{q_v}, \tag{5}$$

where $D_t$ denotes the material derivative: $D_t = \partial_t + \boldsymbol{u} \cdot \nabla$ and $\pi$ is the normalized pressure perturbation. Following Grabowski and Smolarkiewicz (1996), buoyancy is defined as

$$B = g \left[ \frac{\theta - \theta^e}{\theta^r} + \epsilon \left( q_v - q_v^e \right) - \left( q_l - q_l^e \right) \right]. \tag{6}$$

The condensation rate $C$ in eqs. (4) and (5) and the liquid water mixing ratio $q_l$ in eq. (6) come from the Lagrangian microphysics scheme. The terms $F_*$ represent a total forcing due to surface fluxes, radiative heating/cooling, large-scale subsidence and absorbers, while the terms $\mathcal{D}_*$ represent contributions from a SGS turbulence model. The dry-air density is assumed to be equal to the reference state density profile $\rho_d^r$ and, characteristically for the anelastic approximation, the dry-air density at given position does not change with time: $\partial_t \rho_d^r = 0$. By putting $\partial_t \rho_d^r = 0$ into the continuity equation, the following constraint

on the velocity field is obtained:

$$\nabla \cdot (\rho_d^r \boldsymbol{u}) = 0, \tag{7}$$

which will be referred to as the *anelastic constraint*. Throughout the model, the pressure is assumed to be equal to the environmental pressure profile $p^e(z)$. The only exception is the pressure gradient term appearing in eq. (3), in which the pressure is adjusted so that $\boldsymbol{u}$ satisfies the anelastic constraint (eq. 7) (Lipps and Hemler, 1982; Grabowski and Smolarkiewicz, 1996).

UWLCM offers two methods for modeling diffusion of Eulerian variables due to the SGS turbulence. The first is an implicit LES (ILES) approach, in which there is no explicit parametrisation of SGS mixing, i.e. $\mathcal{D}_* \equiv 0$. Instead, numerical diffusion of the advection scheme is used to mimic the SGS turbulence (Grinstein et al., 2007). The MPDATA algorithm is argued to be well-suited for ILES simulations (Margolin and Rider, 2002; Margolin et al., 2006). The other method is a Smagorinsky-type model (Smagorinsky, 1963; Lilly, 1962) with the SGS effects parametrised as

$$\boldsymbol{\mathcal{D}}_u = \frac{1}{\rho_d^r} \nabla \cdot (\rho_d^r K_m \boldsymbol{E}), \tag{8}$$

$$\mathcal{D}_\theta = \frac{1}{\rho_d^r} \nabla \cdot (\rho_d^r K_h \nabla \theta), \tag{9}$$

$$\mathcal{D}_{q_v} = \frac{1}{\rho_d^r} \nabla \cdot (\rho_d^r K_q \nabla q_v), \tag{10}$$

where $K_m$ is the eddy viscosity, $K_h$ and $K_q$ are the eddy diffusivities, and $\boldsymbol{E} = \nabla \boldsymbol{u} + (\nabla \boldsymbol{u})^T - \frac{2}{3}(\nabla \cdot \boldsymbol{u})\boldsymbol{I}$ is the deformation tensor. The eddy viscosity is given by

$$K_m = \begin{cases} (c_s \lambda)^2 |\boldsymbol{E}| \left(1 - \frac{K_h}{K_m} \mathrm{Ri}\right)^{1/2}, & \text{if } \frac{K_h}{K_m} \mathrm{Ri} < 1 \\ 0 & \text{otherwise,} \end{cases} \tag{11}$$

where $c_s$ is the Smagorinsky constant, $\lambda$ is the mixing length, and Ri is the Richardson number. The eddy diffusivities are

$$K_h = K_q = K_m / \mathrm{Pr}, \tag{12}$$

where Pr is the Prandtl number. Following Schmidt and Schumann (1989) the mixing length is set to $\lambda = \min(\Delta, c_L z)$. Given highly anisotropic grid cells used in stratocumulus simulations we set $\Delta = \Delta z$, as in the Colorado State University System for Atmospheric Modeling (Khairoutdinov and Randall, 2003). The values of the numerical constants are taken following Schmidt and Schumann (1989) as $c_s = 0.165$, $\mathrm{Pr} = 0.42$, and $c_L = 0.845$.

## 2.2 Lagrangian particles

Liquid water is modeled with a Lagrangian, particle-based microphysics scheme from the libcloudph++ library (Arabas et al., 2015). It is an implementation of the Super-Droplet Method (SDM) (Shima et al., 2009). The key idea is to represent all liquid particles using a small number of computational particles, called super-droplets (SDs). Each SD represents a large number of real particles. The number of real particles represented by a given SD is called the multiplicity (also known as the weighting

factor), and is denoted by $\xi$. Other attributes of SDs are the dry radius $r_d$, the wet radius $r$, the hygroscopicity parameter $\kappa$ and the position $\boldsymbol{x}$ in the model domain.

The condensational growth rate of a SD is equal to that of a single real particle. We calculate it using the Maxwell-Mason approximation (see Arabas et al. 2015):

$$r\frac{dr}{dt} = \frac{D'_{\text{eff}}}{\rho_w}\left(q_v - q_{vs}a_w\left(r,r_d,\kappa\right)\exp(A/r)\right), \tag{13}$$

where

$$\frac{1}{D'_{\text{eff}}} = (D\rho_d)^{-1} + K^{-1}q_{vs}\frac{l_v}{T}\left(\frac{l_v}{R_vT} - 1\right) \tag{14}$$

and water activity is calculated using the $\kappa$-Köhler parametrisation (Petters and Kreidenweis, 2007):

$$a_w\left(r,r_d,\kappa\right) = \frac{r^3 - r_d^3}{r^3 - r_d^3(1-\kappa)}. \tag{15}$$

Following Lipps and Hemler (1982), the relative humidity is defined as $\phi = q_v/q_{vs}$ and the saturation water vapor mixing ratio is calculated using the formula $q_{vs} = (R_d/R_v)e_s/(p^e - e_s)$. Formulas for $A$ and $l_v$ can be found in Arabas and Pawlowska (2011). The vapor and heat diffusion coefficients $D$ and $K$ include gas kinetic and ventilation effects and are evaluated as in Arabas et al. (2015).

Collision-coalescence of droplets is treated as a stochastic process (Gillespie, 1972). Collisions are possible only between
droplets that are located within the same spatial cell, called the coalescence cell. It is assumed that coalescence cells are well-mixed, i.e. that droplets are randomly and uniformly distributed within a coalescence cell. Then, the probability that any two real droplets $j$ and $k$ that are located in the same coalescence cell coalesce during the time interval $\Delta t_c$ is given by the equation (Shima et al., 2009)

$$P_{j,k} = K_{j,k}\frac{\Delta t_c}{\Delta V}, \tag{16}$$

where $K_{j,k}$ is the collision-coalescence kernel for these two droplets and $\Delta V$ is the volume of the coalescence cell. The probability of coalescence of SDs needs to be increased to account for the fact that each SD represents a large number of real droplets. The probability that any two SDs $j$ and $k$ that are in the same coalescence cell coalesce during the time interval $\Delta t_c$ is related to the probability of coalescence of real droplets in the following manner (Shima et al., 2009):

$$P_{j,k}^{\text{SD}} = \xi_k P_{j,k}, \tag{17}$$

where SDs are labeled so that $\xi_j \leq \xi_k$ and this convention is assumed throughout the rest of this paragraph. Coalescence of the two SDs is interpreted as a coalescence of $\xi_j$ pairs of real droplets. Each pair consists of one real droplet represented by the $j$-th SD and one real droplets represented by the $k$-th SD. The remaining $\xi_k - \xi_j$ real droplets represented by the $k$-th SD are not affected by the coalescence of these two SDs.

Such treatment of coalescence, sometimes referred to as the *all-or-nothing* algorithm, assures that coalescence does not
increase the number of SDs. This algorithm was found to give the best results in a recent comparison of various coalescence

algorithms used in Lagrangian schemes for microphysics (Unterstrasser et al., 2017). Dziekan and Pawlowska (2017) showed that the *all-or-nothing* algorithm produces correct realizations of the stochastic coalescence process described in Gillespie (1972), but only for $\xi = 1$. For $\xi > 1$, an average over realizations of the *all-or-nothing* algorithm is in good agreement with the expected value of the stochastic process, but the variability between realizations is much higher. This is because the number

of SDs is much smaller than the number of real droplets. In consequence, the statistical sample for $\xi > 1$ is much smaller than in the more realistic case of $\xi = 1$. The collision-coalescence algorithm is not the only cause of the high variability for $\xi > 1$. Motion of SDs is also expected to give a high variability, because when a SD moves from one spatial cell to another, a large number of real particles is abruptly moved between these cells. It is not certain if the high variability in the SDM associated with collision-coalescence of SDs and motion of SDs has any impact on averaged properties of a modeled cloud. To determine

if it does have an effect, we conduct simulations for various number of SDs (see section 4.2).

Super-droplets are treated as non-inertial particles that always sediment with their terminal velocity. There is an option to model diffusion of liquid water due to the SGS turbulence by adding a random velocity component $\boldsymbol{u}'_{\mathrm{SD}}$ that is specific to each SD. Each component of this velocity perturbation evolves according to eq. (10) from Grabowski and Abade (2017). It is important to note that this SGS velocity can only be added when the Smagorinsky scheme is used for Eulerian variables.

Altogether, velocity of a SD is equal to $\boldsymbol{u}_{\mathrm{SD}} = \boldsymbol{u} + \boldsymbol{u}'_{\mathrm{SD}} + (0, 0, w_t) + (0, 0, w_{\mathrm{LS}})$. This formula represents the combined effects of transport by the resolved air flow, SGS turbulence, sedimentation and large-scale subsidence.

## 3   Numerical algorithms

### 3.1   Numerical integration of Eulerian equations

Numerical integration of the governing Eulerian equations is done using the MPDATA algorithm implemented in libmp-

data++ (Jaruga et al., 2015). MPDATA is an algorithm for solving the generalized transport equation (Smolarkiewicz, 2006)

$$\partial_t (G\psi) + \nabla \cdot (G\boldsymbol{u}\psi) = GR, \tag{18}$$

where $\psi$ is a scalar field advected by the velocity field $\boldsymbol{u}$, $R$ is the source/sink right-hand side (RHS) and G can represent the fluid density, the Jacobian of coordinate transformation or their product. The equivalent of eq. (18) in Lagrangian description

is:

$$D_t\psi = R. \tag{19}$$

Equation (3) for components of vector $\boldsymbol{u}$ and eqs. (4) and (5) have the same form as eq. (19). Equation (19) introduces notation that is convenient for presenting the numerical integration procedure of UWLCM. All RHS terms, except buoyancy and pressure gradient terms in eq. (3), are integrated with the forward Euler method. These terms are denoted by $R_{\mathrm{E}}$. The

buoyancy and pressure gradient terms, denoted by $R_{\mathrm{T}}$, are applied using the trapezoidal rule. The integration algorithm is:

$$\psi^{[n+1]} = ADV\left(\psi^{[n]} + \Delta t R_{\mathrm{E}}^{[n]} + 0.5\Delta t R_{\mathrm{T}}^{[n]}, \boldsymbol{u}^{[n+1/2]}\right) + 0.5\Delta t R_{\mathrm{T}}^{[n+1]}, \tag{20}$$

where $ADV\left(\psi, \boldsymbol{u}\right)$ is an operator representing MPDATA advection of a scalar field $\psi$ by the velocity field $\boldsymbol{u}$. Superscripts denote the time level. The mid-time-level velocity field $\boldsymbol{u}^{[n+1/2]}$ is obtained by linear extrapolation from $\boldsymbol{u}^{[n-1]}$ and $\boldsymbol{u}^{[n]}$.

Pressure perturbation $\pi$ is adjusted so that the velocity field satisfies eq. (7). By applying eq. (7) to the equation for $\boldsymbol{u}^{[n+1]}$ discretised in the form of eq. (20), the following elliptic equation for $\pi^{[n+1]}$ is obtained:

$$\nabla \cdot \left[\rho_d^r \left(\hat{\boldsymbol{u}} + 0.5\Delta t \boldsymbol{k} B^{[n+1]} - 0.5\Delta t \nabla \pi^{[n+1]}\right)\right] = 0, \tag{21}$$

where

$$\hat{\boldsymbol{u}} = ADV\left[\boldsymbol{u}^{[n]} + \Delta t\left(\boldsymbol{F}_u^{[n]} + \boldsymbol{\mathcal{D}}_u^{[n]}\right) + 0.5\Delta t\left(-\nabla\pi^{[n]} + \boldsymbol{k}B^{[n]}\right), \boldsymbol{u}^{[n+1/2]}\right] \tag{22}$$

and the thermodynamic fields required in $B^{[n+1]}$ are already available when the equation has to be solved. The pressure problem stated in eq. (21) is solved with the generalized conjugate residual solver (Smolarkiewicz and Margolin, 2000; Smolarkiewicz and Szmelter, 2011).

## 3.2 Numerical algorithms for super-droplets

For numerical reasons, condensational growth of SDs is solved in terms of the squared wet radius (Chen, 1992; Shima et al., 2009). Integration of eq. (13) is done with a scheme that is implicit with respect to the wet radius and explicit with respect to $q_v$ and $\theta$:

$$r^{2[n+1]} = r^{2[n]} + \Delta t \left.\frac{dr^2}{dt}\right|_{r^{2[n+1]}, q_v^{[n]}, \theta^{[n]}}. \tag{23}$$

Solution for eq. (23) is found with a predictor-corrector procedure. We refer the reader to Arabas et al. (2015) for details of this procedure. Condensation can rapidly change radii of small droplets. Therefore to correctly model condensation, in particular during the crucial moment of droplet activation, it is necessary to model condensation with a relatively short time step. Tests performed in a kinematic 2D model of stratocumulus clouds have shown that the number of activated droplets converges for condensation time step of around $0.1$s. A typical time step $\Delta t$ of a LES model is around $1$s. Therefore it is necessary to do several condensation time steps in a single LES time step, a procedure we call substepping. To explain the idea of the substepping algorithm, we introduce the following notation: $S_c$ for the number of substeps, $\boldsymbol{\psi} = (\theta, q_v)$ for a vector of Eulerian variables, $\boldsymbol{\psi}_{\mathrm{old}}$ for values of Eulerian variables after the substepping algorithm finished in the previous time step and $\boldsymbol{\psi}_{\mathrm{new}}$ for values of Eulerian variables before the start of the substepping algorithm in the current time step. In the first substep, Eulerian variables are set to $\boldsymbol{\psi}_{\mathrm{old}} + \frac{\boldsymbol{\psi}_{\mathrm{new}} - \boldsymbol{\psi}_{\mathrm{old}}}{S_c}$ and then condensation is calculated using the procedure defined in eq. (23). Please note that this condensation procedure changes Eulerian variables. In each subsequent time step, $\frac{\boldsymbol{\psi}_{\mathrm{new}} - \boldsymbol{\psi}_{\mathrm{old}}}{S_c}$ is added to Eulerian variables and then the condensation procedure is run again. Two types of the substepping algorithm are considered that differ only in the spatial cell from which the value of $\boldsymbol{\psi}_{\mathrm{old}}$ is diagnosed. In the *per-particle* algorithm, $\boldsymbol{\psi}_{\mathrm{old}}$ is diagnosed from the cell in which the given SD was in the *previous* time step. In the *per-cell* algorithm, $\boldsymbol{\psi}_{\mathrm{old}}$ is diagnosed from the cell in which given SD is in the *current* time step. The *per-cell* algorithm is computationally less demanding, because $\boldsymbol{\psi}_{\mathrm{old}}$ is the same for all SDs in a given spatial cell. In the *per-particle* algorithm $\boldsymbol{\psi}_{\mathrm{old}}$ can be different for different SDs in the same cell, so each

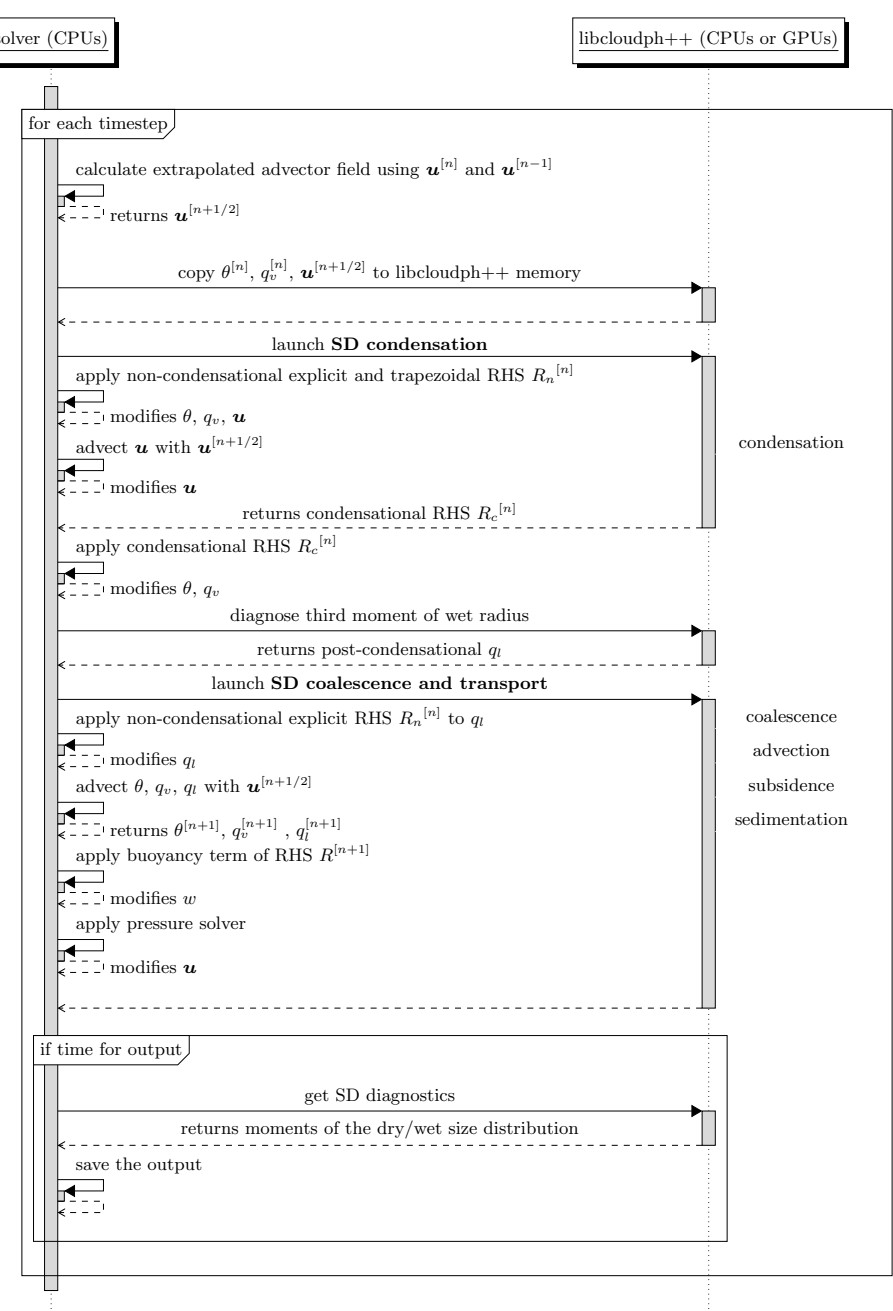

**Figure 1.** UML sequence diagram showing the order of operations within a single time step. Calls in boldface start microphysical calculations that are done on GPUs simultaneously with solver operations done on CPUs. The RHS is divided into condensational and non-condensational parts, $R = R_n + R_c$.

SD needs to remember its own value of $\psi_{\text{old}}$. Moreover, in the *per-particle* algorithm, values of pressure and density also need to vary between substeps. This is not necessary in the *per-cell* algorithm, because pressure and density in a given cell are constant in time. A more detailed description of the substepping algorithms and a comparison of results they produce is given in appendix B. The conclusion from that comparison is that the *per-cell* algorithm is correct for stationary clouds, but gives significant errors if cloud edge moves, while the *per-particle* algorithm is correct in both cases. All presented results of modeling stratocumulus clouds were obtained using the *per-cell* algorithm.

The stochastic collision-coalescence process described in section 2.2 is modeled with a Monte Carlo algorithm developed by Shima et al. (2009). The key feature of this algorithm is that each SD can collide only with one other SD during a time step. Thanks to that, computational cost of the algorithm scales linearly, and not quadratically, with the number of SDs. Dziekan and Pawlowska (2017) showed that this "linear sampling" technique does not affect the mean, nor the standard deviation, of the results. Note that in the coalescence algorithm of Shima et al. (2009), the same pair of SDs can collide multiple times during one time step. This feature was not implemented in libcloudph++ at the time when the paper Arabas et al. (2015) was published. libcloudph++ has been modified since then and now multiple collisions are allowed. It is possible to run the coalescence algorithm with a shorter time step than the model time step. Then, coalescence is calculated more than once in each model time step, a procedure we call coalescence substepping.

The procedure for initialization of SD sizes is described in detail in Dziekan and Pawlowska (2017), where it is called the "constant SD" initialization. In short, the range of initial values of $r_d$ is divided into $N_{\text{SD}}$ bins, which have the same size in $\log(r_d)$. In each bin, a single value or dry radius is randomly selected and assigned to a single SD. Multiplicity of the SD is readily calculated from the initial aerosol size spectrum. Next, wet radius is initialized to be in equilibrium with the initial relative humidity. If the initial relative humidity is higher than $0.95$, the wet radii are initialized as if it was equal to $0.95$. This procedure is performed for each spatial cell. This initialization algorithm gives a good representation of the initial size spectrum even for small values of $N_{\text{SD}}$.

Advection of SDs is modeled with a predictor-corrector algorithm described in Grabowski et al. (2018a). Simpler, first-order algorithms for advection were found to cause inhomogeneous spatial distributions of SDs, with less SDs in regions of high vorticity.

## 3.3 Order of operations

The sequence of operations done in a single time step is presented on a Unified Modeling Language (UML) sequence diagram in fig. 1. The diagram is a convenient way of showing how coupling between Eulerian dynamics and Lagrangian microphysics is done. The diagram also shows operations that are done simultaneously on CPUs and GPUs. Please note how the liquid water mixing ratio $q_l$ is treated. Liquid water is resolved by the SDM and $q_l$ could be diagnosed from the super-droplet size spectrum each time it is needed in the buoyancy term in eq. (3) or radiative term in eq. (4). Buoyancy is integrated with a trapezoidal scheme, which requires $q_l$ after advection to be known. In a straightforward implementation, in which $q_l$ is diagnosed from SDs after advection of SDs, pressure solver calculations can only be started after advection of SDs has been calculated. Then, there is little parallelism of calculations on GPUs and CPUs. To achieve more parallelism, we introduce an auxiliary Eulerian

field for $q_l$. Value of $q_l$ is diagnosed from SDs once per time step, after condensation calculation. Then, $q_l$ advection is done using a first-order accurate upwind scheme. Using the auxiliary $q_l$ field, it is possible to calculate coalescence and motion of SDs simultaneously with calculations of advection of Eulerian fields and of the pressure problem.

### 3.4 Spatial discretisation

5   Eulerian dependent variables of the model are co-located. Their positions form the nodes of the primary grid. However, the libmpdata++ advection algorithms are formulated using a dual, staggered Arakawa-C grid (Arakawa and Lamb, 1977). The cell centers of the dual grid are the nodes of the primary mesh. Schematic of a 2D computational domain with the Arakawa-C grid is shown in fig. 2. Throughout this paper, by "grid cells", "Eulerian cells" or simply "cells", we refer to the cells of the dual grid. To form the Arakawa-C arrangement, components of the vector $\boldsymbol{u}$ are linearly interpolated to the edges of the

10   dual grid (see Jaruga et al. (2015) for details). Super-droplets are restricted to the physical space, which is the shaded region in fig. 2. Coupling of Eulerian variables with SDs is done using the dual grid. All SDs that are located in the same cell of the dual grid are subjected to the same conditions that are equal to the values of scalars residing at the center of the cell. Similarly, condensation of a given SD affects scalars in the center of the dual grid cell, in which this SD is located. To calculate velocity of air that advects a given SD, velocities, which reside at the edges of the dual grid, are interpolated to the position

15   of the SD. The interpolation is done linearly, separately in each dimension, as advocated by Grabowski et al. (2018a). Spatial discretisation is also necessary in the algorithm for modeling collision-coalescence (cf. section 2.2). We use the dual grid cells also as coalescence cells, with the exception of the cells at the domain edges. There, only the physical (shaded) part of dual grid cells is used as coalescence cells.

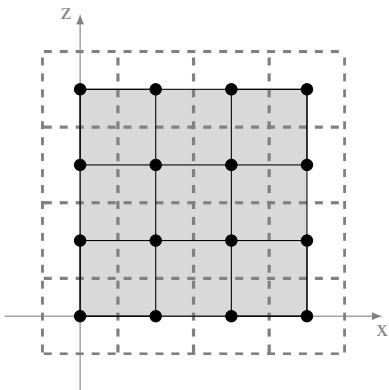

**Figure 2.** Schematic of a 2D computational domain. Bullets mark the data points for the dependent variable $\psi$ in eq. (18), solid lines depict edges of primary grid and dashed lines mark edges of dual grid. Reproduced from Jaruga et al. (2015).

## 4 Comparison with other models - marine stratocumulus simulations

UWLCM simulations of a marine stratocumulus cloud are presented in this section. The main goal is to validate UWLCM by comparing it with other LES models. We study sensitivity of the model to the way the SGS turbulence is modeled and to values of the microphysics scheme parameters. Results of this sensitivity study may provide some guidance to other users of Lagrangian microphysical schemes. The simulation setup is based on observations made during the second Dynamics and Chemistry of Marine Stratocumulus (DYCOMS-II) field study (Stevens et al., 2003). The setup, described in detail in Ackerman et al. (2009), is an idealization of conditions observed during the second research flight (RF02) of this campaign. Both heavily drizzling open cells and lightly drizzling closed cells were sampled by RF02. The initial thermodynamic conditions are an average from both types of cells and the microphysical conditions are an average over heavily drizzling cells only. Comparison of simulation results from 11 different LES models is presented in Ackerman et al. (2009). There is a large variability in the amount of drizzle predicted by different models, which illustrates how difficult it is for LES models to reproduce precipitation formation. One of the reasons why we chose to test UWLCM using this setup is to test how well Lagrangian microphysics performs in modeling drizzle. The models that took part in the intercomparison use either bin microphysics (one model with single-moment bin and one model with double-moment bin) or bulk microphysics (2 models with single-moment bulk and 7 models with double-moment bulk).

### 4.1 Simulation setup

The simulation setup follows Ackerman et al. (2009). The domain size is 6.4km x 6.4km x 1.5km with a regular grid of cells of 50m x 50m x 5m size. Rigid and periodic boundary conditions are used at the vertical and horizontal edges of the domain, respectively. The simulations are run for 6 hours. The initial profiles of $q_v$ and $\theta$ give high values of supersaturation in the layer in which a cloud was observed. However, the simulation is initialized without any cloud water, because it is not known analytically what should be the initial wet radius distribution. First part of the simulation, called the *spinup* period, is dedicated to obtaining a stationary distribution of wet radii. During the spinup, the collision-coalescence process is turned off and the supersaturation in the condensational growth equation is limited to 1%. Please note that in Ackerman et al. (2009) this supersaturation limit is applied only to the activation and not to the condensational growth. This approach can not be used in UWLCM, because in UWLCM activation is not modeled as a separate process. The spinup period is 1 hour long, which was found to be long enough to reach a stationary concentration of cloud droplets – indicating a stationary spectrum of the wet radius. Aerosol is assumed to consist of ammonium sulfate with the initial size distribution as defined in Appendix A of Ackerman et al. (2009). Following Petters and Kreidenweis (2007), the hygroscopicity parameter for ammonium sulfate is $\kappa = 0.61$. Collision efficiencies are taken from Hall (1980) for large droplets and from Davis (1972) for small droplets. Coalescence efficiency is set to 1. Terminal velocities are calculated using a formula from Khvorostyanov and Curry (2002). libmpdata++ allows the user to choose from a number of MPDATA options. In the presented simulations, we use the "infinite-gauge" option *iga* for handling variable-signed fields combined with the non-oscillatory option *fct*.

## 4.2 Two-dimensional simulations: sensitivity study of SDM

Two-dimensional simulations are used to investigate sensitivity of the model to parameters of the SDM: the coalescence time step length $\Delta t_{\mathrm{coal}}$ and the initial number of SDs $N_{\mathrm{SD}}$. Results are compared with 3D simulations from the Ackerman et al. (2009) in order to assert if 2D simulations, which are computationally cheap, give reasonable representation of some of the features of 3D simulations. However, it has to be kept in mind that the turbulence behavior in 2D is fundamentally different from 3D. Simulations are run for two model time step lengths, $\Delta t = 0.1$s and $\Delta t = 1$s. No substepping is done for $\Delta t = 0.1$s. Results of this simulations provide a reference for simulations with longer time steps. In simulations with $\Delta t = 1$s, 10 substeps for condensation are done, hence condensation time step is $\Delta t_{\mathrm{cond}} = 0.1$s. Using a longer condensation time step results in activation of too many aerosols (result not shown in the following figures for clarity). Two values of the coalescence time step are tested: $\Delta t_{\mathrm{coal}} = 1$s (no coalescence substepping) and $\Delta t_{\mathrm{coal}} = 0.1$s (10 coalescence substeps) in combination with two values of the initial number of SDs: $N_{\mathrm{SD}} = 40$ (which is used in 3D simulations) and $N_{\mathrm{SD}} = 1000$. Since the goal of 2D simulations is to study the microphysical model, SGS turbulence is modeled using the ILES approach. In 2D, we observe significant variability in results of simulation runs done for the same parameter values. The variability comes from two sources. One is that the initial thermodynamic conditions include a small random perturbation. The other is that initialization of SD radii and collision-coalescence of SDs are modeled with Monte Carlo algorithms. To compensate for this inherent variability, all shown UWLCM results of 2D simulations are averages from ensembles of 10 simulations.

Time series of selected domain averaged variables are shown in fig. 3. The only significant relationship between model parameters and results is that the amount of surface precipitation is ca. two times higher for $\Delta t_{\mathrm{coal}} = 1$ s than for $\Delta t_{\mathrm{coal}} = 0.1$ s. Stronger precipitation induces differences in LWP that, up to the onset of precipitation, is the same for all parameter combinations. The amount of surface precipitation does not depend on the model time step nor on the number of SDs. Interestingly, 2D simulations show an abrupt increase in the entrainment rate and in the maximum of variance of $w$ around 3h of the simulations. This increase is preceded by the moment when first precipitation reaches the surface. This suggests that the increase in the maximum of variance of $w$ and in the entrainment rate is caused by rain evaporation. The need for the spinup period for microphysics is best seen on the $N_c$ time series. Initially, due to the large initial supersaturation, cloud droplets form on all aerosol particles. Afterwards, $N_c$ quickly decreases and after 1h reaches the value of ca. 60 $\mathrm{cm}^{-3}$, in agreement with the 3D reference simulations.

Vertical profiles from the 2D simulations are shown in fig. 4. As already observed in time series plots, precipitation flux strongly depends on $\Delta t_{\mathrm{coal}}$. Precipitation flux profile reveals that precipitation flux also weakly depends on $N_{\mathrm{SD}}$; it is slightly lower for $N_{\mathrm{SD}} = 1000$ than for $N_{\mathrm{SD}} = 40$. A similar observation was made in Dziekan and Pawlowska (2017), where the autoconversion efficiency was shown to increase with $N_{\mathrm{SD}}$. The most striking differences between the 2D UWLCM and 3D reference simulations are seen on the profiles of moments of the vertical velocity distribution. This is associated with the decreased dimensionality of our simulations. Interestingly, profiles of VAR($w$) and of the third moment of $w$ are in better agreement with observations (see fig. 3 in Ackerman et al. (2009)) in the 2D UWLCM than in the 3D reference simulations.

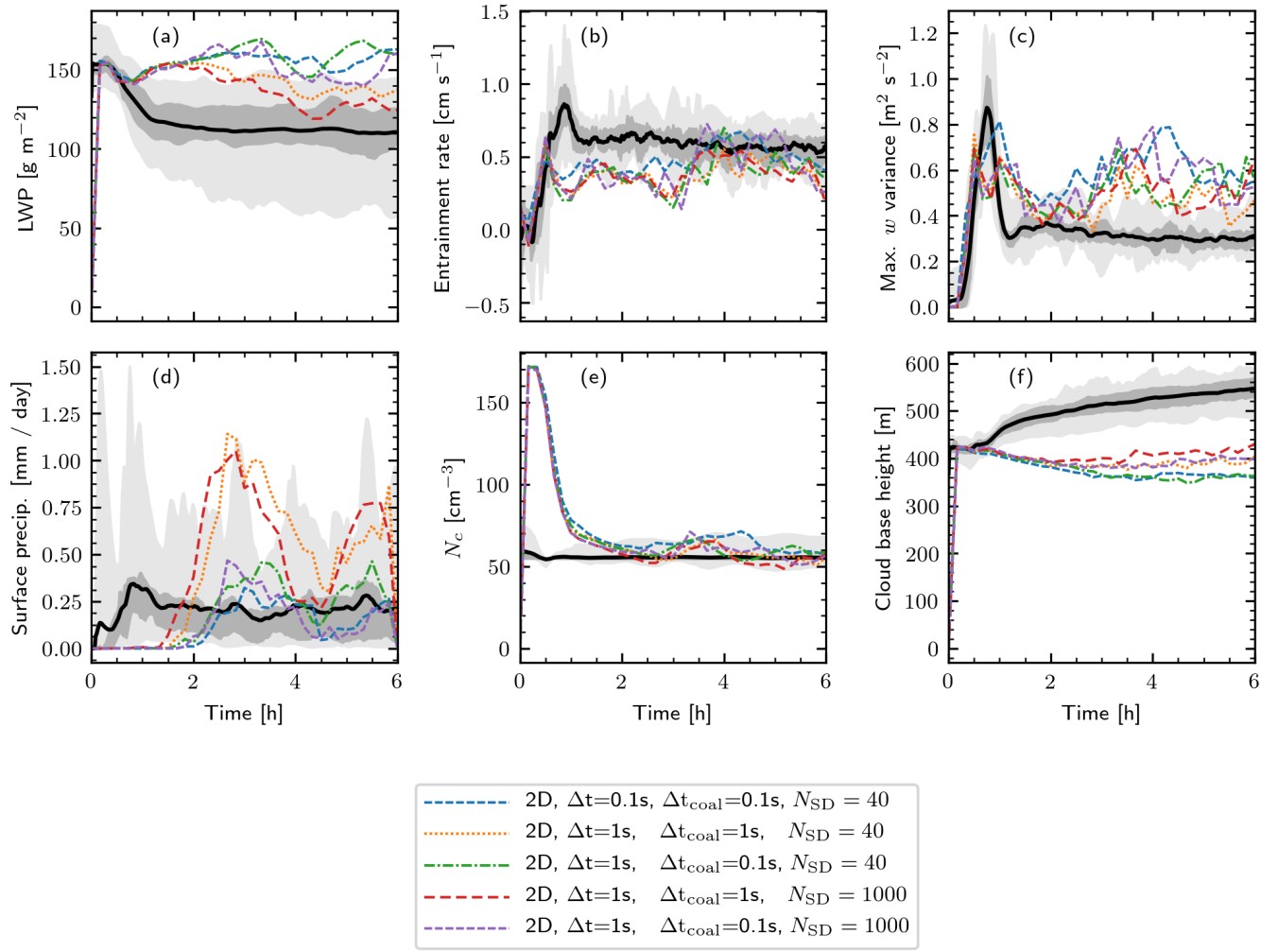

**Figure 3.** 2D UWLCM results. Time series of the domain averaged liquid water path, entrainment rate (equal to $dz_i/dt + w_{LS}z_i$), vertical velocity variance VAR($w$) maximum, surface precipitation, concentration of cloud droplets in cloudy cells and cloud base height. UWLCM simulations were done for different model and coalescence time steps and different initial numbers of SDs per cell. Each colored line represents an average from 10 UWLCM simulations of a given type. Results of 3D simulations from an ensemble of 11 models are shown for reference (Ackerman et al., 2009). Mean, middle two quartiles, and range of that reference ensemble are plotted with the black solid line, the dark shaded region and the light shaded region, respectively.

A conclusion for SDM modeling is that coalescence needs to be resolved with a time step of the order of 0.1 s, although more rigorous convergence tests should be carried out in the future. This conclusion is surprising, because coalescence tests of SDM in box models give correct results for time steps larger than 1s (result not shown) and Shima et al. (2009) estimated that the coalescence algorithm should work well for $\Delta t_{\mathrm{coal}}$ of the order of 1 s. Moreover, one might expect that using large $\Delta t_{\mathrm{coal}}$ should give too little precipitation, as large $\Delta t_{\mathrm{coal}}$ can make the mean number of collisions to be lower than expected. This is because SDM handles large $\Delta t_{\mathrm{coal}}$ by allowing multiple collisions between SDs and sometimes, when one of the SDs has low multiplicity, not all of these multiple collisions can be realized. However, we see that surface precipitation increases with $\Delta t_{\mathrm{coal}}$. A possible explanation is that for large $\Delta t_{\mathrm{coal}}$ some of the SDs become extremely lucky and grow much faster than expected due to multiple collisions. Then, even if the mean number of collisions is lower than it should be, some SDs become very large and cause the observed high surface precipitation. The second conclusion for SDM modeling that can be drawn from the sensitivity test is that $N_{\mathrm{SD}}$ of the order of 40 is sufficient to obtain correct domain averaged results. Certainly, this does not mean that this relatively low number of SDs is sufficient in all cases. For example, larger number of SDs would probably be needed in simulations in which SDs have more attributes, e.g. when modeling aqueous chemistry. Also, we expect that observables other than domain averages, e.g. related to the spatial structure of a cloud, are more sensitive to the number of SDs. Schwenkel et al. (2018) present in more detail how cloud structure depends on the number of SDs. In general, 2D UWLCM results do not deviate very much from the 3D simulations from Ackerman et al. (2009). The biggest difference is in the cloud base height – cloud layer is significantly deeper in 2D. This shows that cheap 2D simulations can be used to coarsely study microphysical effects in stratocumulus clouds.

### 4.3 Three-dimensional simulations: model validation and SGS effects

Based on conclusions of the 2D sensitivity test, 3D simulations are done for $N_{\mathrm{SD}} = 40$, $\Delta t = 1$ s, $\Delta t_{\mathrm{cond}} = 0.1$ s and $\Delta t_{\mathrm{coal}} = 0.1$ s. Three different models of SGS turbulence are tested: implicit LES, the Smagorinsky scheme and the Smagorinsky scheme with turbulent SGS motion of SDs. Contrary to the 2D simulations, the 3D simulations show very little variability between realizations, thanks to the larger simulation domain. Therefore averaging over an ensemble of simulations, which was necessary in the 2D case, is not needed here and results shown come from single simulation runs. Time series of the results are shown in fig. 5. The biggest difference between different descriptions of the SGS turbulence is in the liquid water content. In ILES, in which there is no diffusion of liquid water, LWP increases over time and is much higher than in the reference simulations. Using the Smagorinsky scheme alone, i.e. without diffusion of liquid water, gives less liquid water than ILES. This indicates that diffusion of Eulerian variables in simulations with the Smagorinsky scheme is higher than in ILES. Still, LWP in that case is close the maximum from the reference models. Using the Smagorinsky scheme with turbulent motion of SDs, which models SGS diffusion of liquid water, further decreases LWP and results in better agreement with the reference models. Using the Smagorinsky scheme gives the best agreement with the reference models also in other variables: entrainment rate, maximum of $\mathrm{VAR}(w)$ and cloud base height. Diffusion of liquid water has visible impact on LWP and therefore needs to be included in Lagrangian microphysics models. Unfortunately, this can not be done in ILES with SDM, because in that case a measure of the SGS energy dissipation is not readily available. Aside from decreasing LWP, SGS diffusion of liquid water is seen to decrease

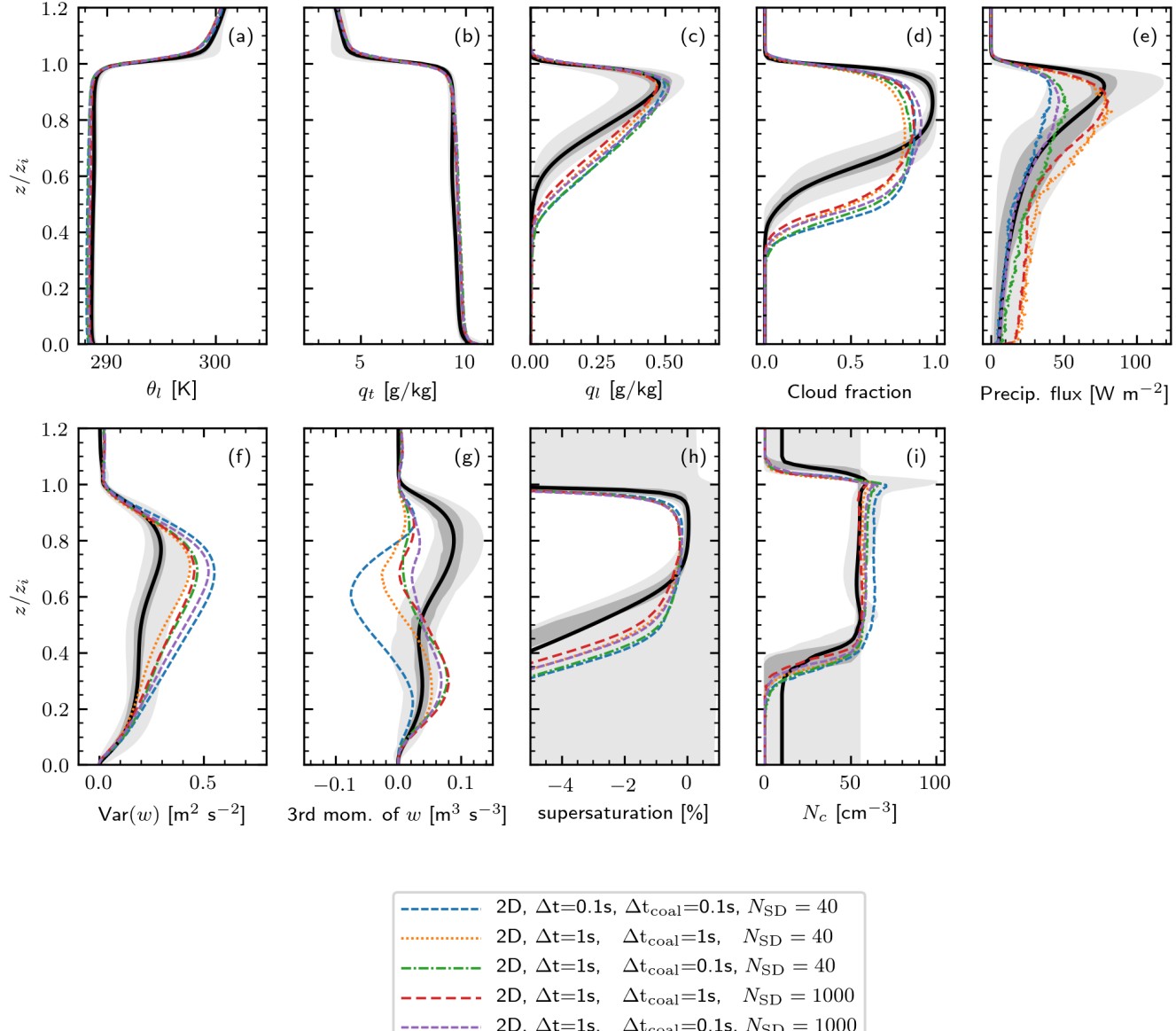

**Figure 4.** 2D UWLCM results. As in fig. 3, but showing horizontally-averaged profiles of liquid water potential temperature (defined in Ackerman et al. (2009)), total water mixing ratio, liquid water mixing ratio, cloud fraction (defined in appendix A), precipitation flux (defined in appendix A), variance of vertical velocity, third moment of vertical velocity, supersaturation and concentration of droplets in cloudy cells. Vertical axis is altitude normalized by inversion height. The profiles are averaged over the 2h to 6h period.

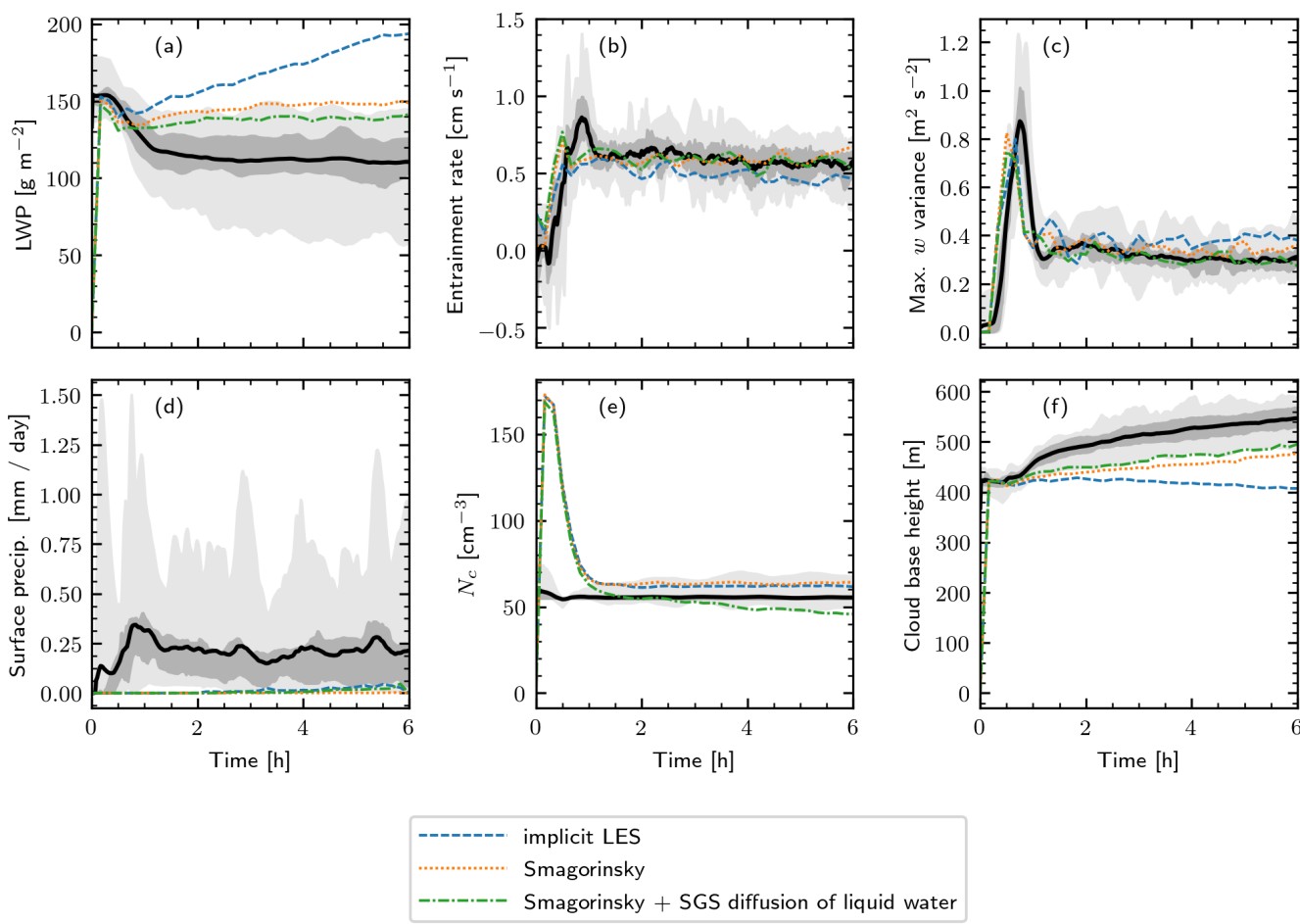

**Figure 5.** As in fig. 3, but for 3D UWLCM simulations. No averaging over ensembles is done, i.e. each line comes from a single UWLCM run.

$N_c$, giving better agreement with the reference models right after the spinup period. Afterwards, SGS diffusion of liquid water causes $N_c$ to slowly decrease with time. A possible explanation is that in regions with little vertical motion, cloud droplets that diffuse out of supersaturated cells will evaporate, but aerosol particles that diffuse into supersaturated cells will not necessarily be activated, because condensational growth of larger cloud droplets already present in this region may consume all available
5   supersaturation. Surface precipitation is very low in all 3D UWLCM simulations. 2D UWLCM simulations give larger surface precipitation than 3D UWLCM, which is attributed to a deeper cloud layer in 2D. There is a very large spread in surface precipitation in the reference models, with some of them producing as little as 3D UWLCM. The subject of discrepancy in surface precipitation is discussed in more detail in Sec. 4.4.

Profiles from 3D simulations are presented in fig. 6. Using the Smagorinsky scheme with SGS diffusion of liquid water
10   gives the best agreement with the reference models, with the exception cloud fraction that is smaller than reference. However,

cloud fraction profile strongly depends on definition of cloudy cells. Following Ackerman et al. (2009), we define cloudy cells as those with concentration of cloud droplets greater than 20 $cm^{-3}$. On the other hand, most of the reference models use parametrised microphysics. Therefore these models define cloudy cells as saturated cells. Using this definition, all UWLCM runs give maximum cloud fraction of ca. 95 %, in agreement with the reference models. Also, cloud cover, defined as fraction

of columns with LWP > 20 $g\,m^{-2}$, is close to 100 % in all 3D UWLCM simulations. Choice of the SGS diffusion model also affects structure of the velocity field. Increasing SGS diffusion strength decreases variance of $w$ and increases skewness of $w$, which shifts from negative for ILES to positive for the Smagorinsky scheme with diffusion of liquid water.

## 4.4   Precipitation results

The purpose of this section is to study discrepancy between the amount of surface precipitation observed during the DYCOMS-

II campaign (from ca. $0.25\,mm/day$ to ca. $0.45\,mm/day$, Ackerman et al. (2009)) and modeled by 3D UWLCM (almost none). Precipitation flux in UWLCM is ca. two times lower than the average of reference simulations (cf. fig. 6). To better understand this issue we make a comparison with the only models with bin microphysics that took part in the reference intercomparison: Distributed Hydrodynamic Aerosol and Radiative Modeling Application (DHARMA, Stevens et al. (2002)) and Regional Atmospheric Modeling System (RAMS, RAMS Technical Description). DHARMA uses single-moment bin

microphysics and RAMS uses double-moment bin microphysics. More details about DHARMA and RAMS simulations of the DYCOMS RF02 case can be found in Appendix B of Ackerman et al. (2009). We compare our results only with these two models, because, contrary to the bulk schemes, bin schemes explicitly resolve size spectrum of droplets and do not rely on parametrisations of the collision-coalescence process, i.e. they are at a similar level of precision as the SDM. Bin microphysics are troubled by artificial broadening of the size spectrum of droplets due to numerical diffusion associated with advection in the

physical space (Morrison et al., 2018). Such artificial broadening increases the rate of collision-coalescence, hence models with bin microphysics might produce too much precipitation. Lagrangian, particle-based schemes such as SDM have no numerical diffusion in the size spectrum.

Time series and profiles showing the amount of liquid water, surface precipitation and concentration of cloud droplets from UWLCM, DHARMA and RAMS are plotted in fig. 7. Precipitation flux and surface precipitation are similar in UWLCM and

RAMS. Both models produce almost no surface precipitation, aside from a short period at the start of the RAMS simulation, when the simulation has not yet reached a stationary state. The DHARMA model stands out in that the amount of surface precipitation it produces is higher, in agreement with observations. Cloud depth, LWP and $N_c$ are similar in DHARMA and UWLCM. So why does DHARMA give a much higher precipitation flux? One possible explanation is that it is a result of artificial broadening of size spectra caused by numerical diffusion. On the other hand, why does UWLCM give less precipitation

than observed? Possibly, precipitation is affected by some physical processes that are not currently modeled by UWLCM. The list of such unresolved processes that could affect precipitation includes, but is not limited to, the following: SGS turbulence affecting condensation and coalescence of droplets, lucky droplets effect and giant CCN initiating rain formation.

In bin microphysics of RAMS and DHARMA, water droplets are artificially divided into haze particles and cloud droplets, and droplet activation is modeled as an instantaneous process (Stevens et al., 1996; Ackerman et al., 1995). Therefore even

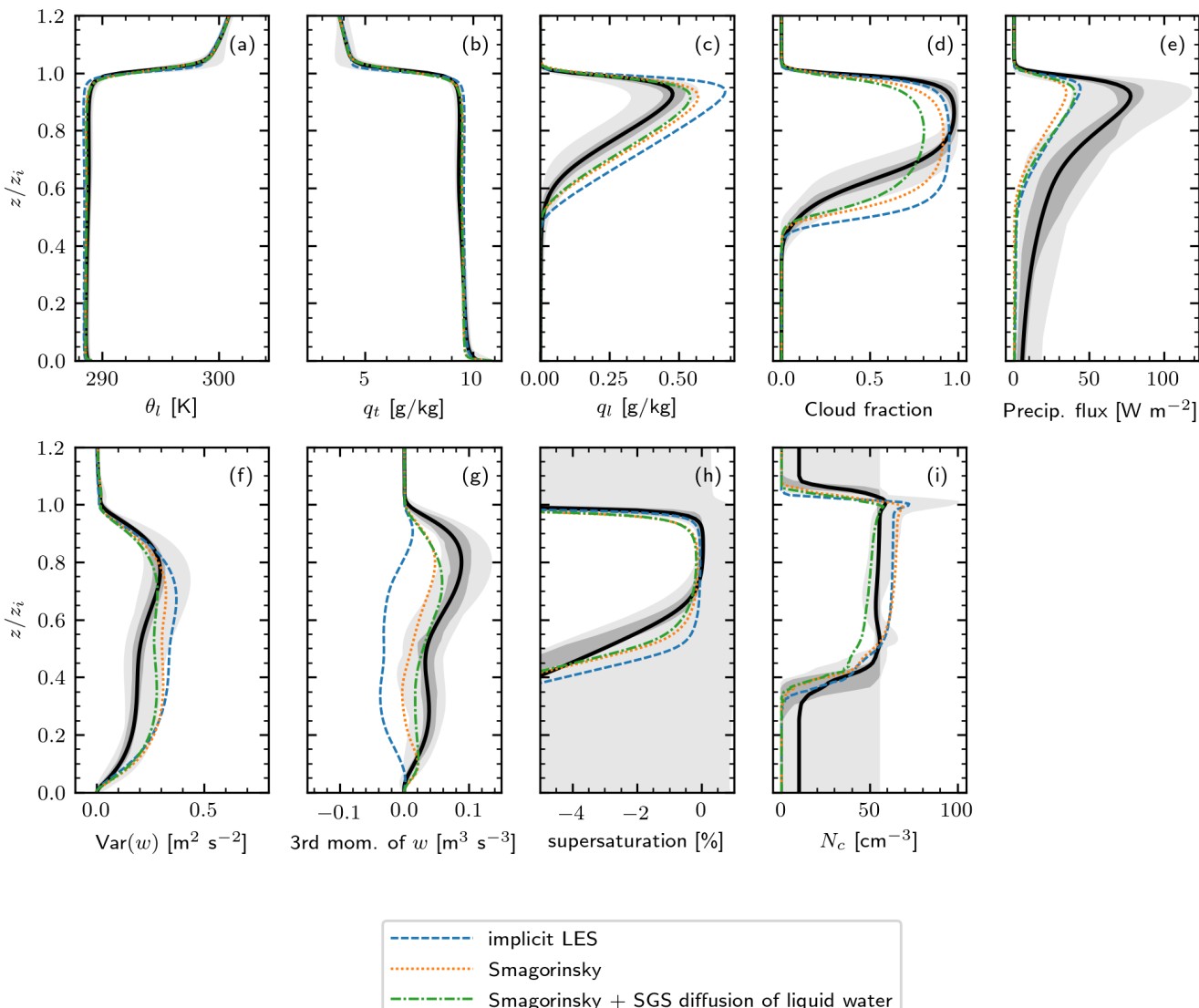

**Figure 6.** As in fig. 4, but for 3D UWLCM simulations. No averaging over ensembles is done, i.e. each line comes from a single UWLCM run.

a short-lived maximum of supersaturation results in activation of new droplets (Hoffmann, 2016). In contrast, in Lagrangian microphysics of UWLCM all water droplets grow according to the same equations, which include curvature and solute terms, and droplet activation is not modeled as a separate process. Thanks to that activation is not instantaneous, but happens over some time. Differences between treatment of activation in bin and Lagrangian microphysics are apparent in the profile of $N_c$
in fig. 7. DHARMA and RAMS predict local maxima of $N_c$ near the cloud base, where supersaturation is highest. In UWLCM, thanks to the explicit treatment of activation, the time scale of activation is resolved and the local maximum of supersaturation near cloud base does not cause activation of new droplets. Therefore $N_c$ in UWLCM monotonously increases near the cloud base.

## 5    Summary

We presented University of Warsaw Lagrangian Cloud Model (UWLCM), a new large-eddy simulations model with Lagrangian particle-based cloud microphysics. The model is built by combining two open-source libraries, one for handling Eulerian dynamics and the other implementing the Lagrangian microphysics scheme. Methods for coupling Lagrangian microphysics with Eulerian dynamics were presented, including spatial discretisation, substepping algorithms and an algorithm for simultaneous computations of Eulerian and Lagrangian components. Simulations of a marine stratocumulus show that the model gives re-
sults in agreement with reference results from 11 other LES models, which proves the capability of Lagrangian microphysics to model realistic clouds. Two-dimensional and three-dimensional simulations of the stratocumulus have been performed. The two-dimensional simulations with UWLCM give reasonable results regarding microphysical phenomena at a fraction of the computational cost of the three-dimensional simulations, and were used to study sensitivity of the Lagrangian microphysics scheme. It was found that the condensational and collisional growth of droplets has to be modeled with a 0.1 s time step and that
number of computational particles does not affect domain averages, apart from small changes in precipitation flux. The 0.1 s time step for condensation and coalescence is realized by doing multiple time steps for these processes in each model time step. Different approaches to modeling SGS turbulence were compared in three-dimensional simulations. The best agreement with other models is obtained by using the Smagorinsky scheme and an algorithm for SGS turbulent motion of computational particles. The implicit LES approach is troubled by the lack of diffusion of liquid water represented by Lagrangian computational
particles. Surface precipitation modeled by UWLCM is lower than observed. This suggests that some physical phenomena not modeled by UWLCM, such as SGS turbulence affecting condensation and coalescence of droplets or giant CCN, are important for precipitation formation. In UWLCM, all particles, including humidified aerosols, evolve according to the same set of equations. Therefore it is not necessary to include droplet activation as an additional process. Advantages of such approach are most apparent near the cloud base, where bin schemes produce local maxima of cloud droplet concentration, while in UWLCM
cloud droplet concentration increases monotonously.

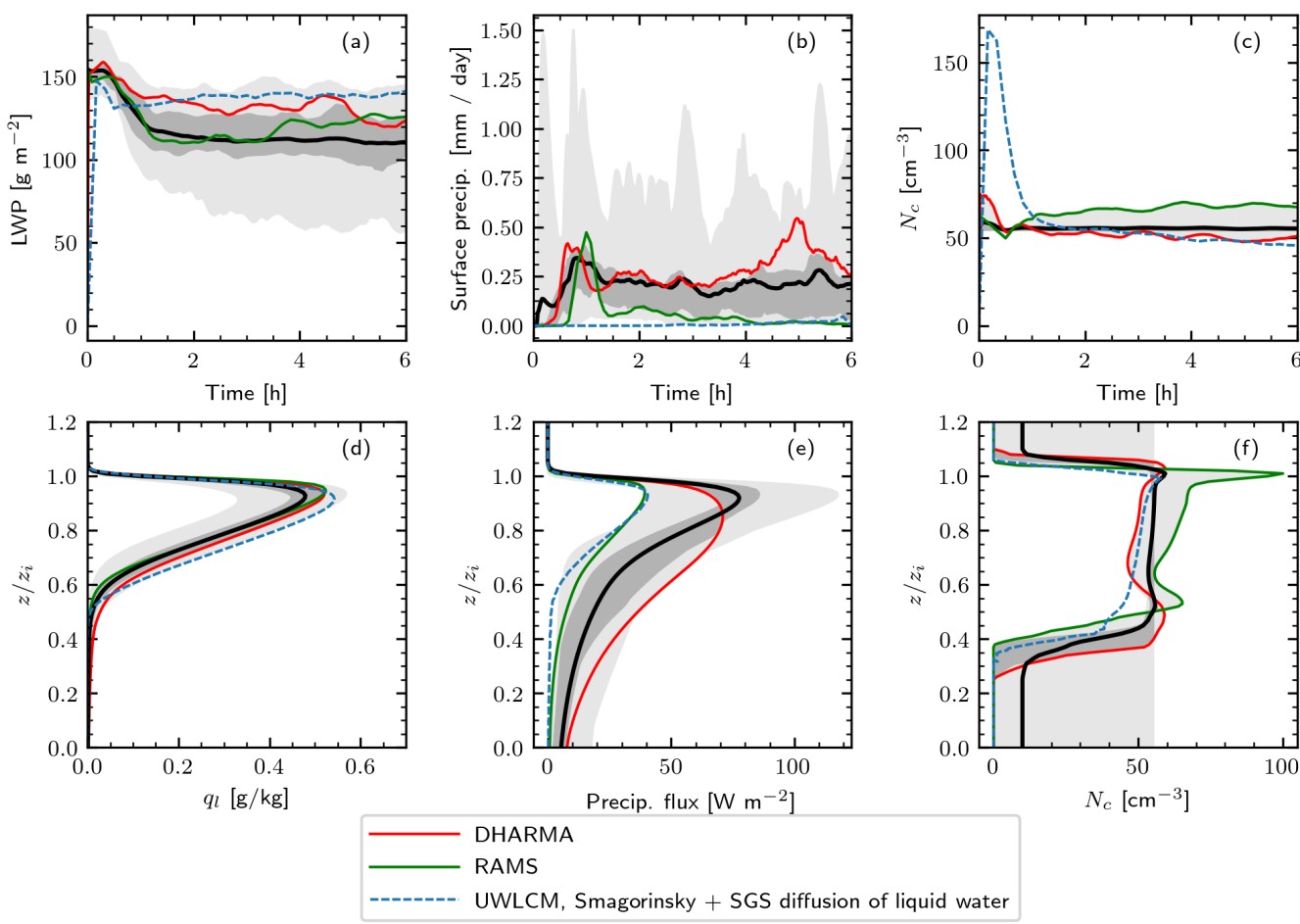

**Figure 7.** Selected time series of domain averages (upper row) and vertical profiles (lower row) from the 3D UWLCM and the two models with bin microphysics that took part in the Ackerman et al. (2009) intercomparison: DHARMA and RAMS. Out of the two DHARMA runs done for the intercomparison, the DHARMA_BO run is shown, because it uses coalescence efficiency closer to unity, that is the value used in UWLCM and RAMS. Profiles are averaged and scaled as in fig. 4. Reference results from all models discussed in Ackerman et al. (2009) are depicted as in fig. 4.

## 6   Code availability

UWLCM, libmpdata++ and libcloudph++ source codes are available at https://github.com/igfuw. In the study, the following code versions were used: UWLCM v1.0 (https://doi.org/10.5281/zenodo.2791156), libmpdata++ v1.2.0 (https://doi.org/10.5281/zenodo.2787740) and libcloudph++ v2.1.0 (https://doi.org/10.5281/zenodo.2790277).

5   ## 7   Data availability

For simulation results, please contact P. Dziekan.

## Appendix A:  List of symbols

**Table A1.** List of symbols. As in Ackerman et al. (2009), cloudy cells are those with concentration of cloud droplets greater than 20 cm$^{-3}$. Cloud droplets are liquid particles with radius in the range 0.5 μm $< r <$ 25 μm. Cloud fraction is the ratio of cloudy cells to the total number of cells. Precipitation flux in a cell is calculated as $\left(\sum \xi \frac{4}{3}\pi r^3 w_t\right)\rho_w l_v/V$, where $V$ is volume of the grid cell and the sum is done over all SDs in the cell.

| Symbol | SI unit | Description |
|---|---|---|
| $\theta = T(p_{1000}/p)^{\frac{R_d}{c_{pd}}}$ | [K] | potential temperature |
| $p_{1000} = 10^5$ | [Pa] | reference pressure |
| $\theta_v, \theta_l = (p_{1000}/p)^{\frac{R_d}{c_{pd}}}\left(T - l_{v0}\frac{q_l}{c_{pd}}\right)$ | [K] | virtual/liquid-water potential temperature |
| $R_d, R_v$ | [J K$^{-1}$ kg$^{-1}$] | gas constants for dry air/water vapor |
| $c_{pd} = 1005$ | [J K$^{-1}$ kg$^{-1}$] | specific heat at const. pressure for dry air |
| $l_v(T)$ | [J kg$^{-1}$] | latent heat of evaporation (cf. Arabas et al. (2015)) |
| $l_{v0} = 2.5 \times 10^6$ | [J kg$^{-1}$] | latent heat of evaporation at the triple point |
| $q_v = m_v/m_d, q_{vs}$ | [kg kg$^{-1}$] | water vapor mixing ratio/saturation vapor mixing ratio |
| $q_l, q_t = q_v + q_l$ | [kg kg$^{-1}$] | liquid-water/total water mixing ratio |
| $m_v, m_d$ | [kg] | mass of water vapor / dry air |
| $\boldsymbol{x} = (x, y, z)$ | [m] | Cartesian coordinates |
| $\boldsymbol{u} = (u, v, w)$ | [m s$^{-1}$] | velocity field in Cartesian coordinates |
| $\pi = (p - p^e)/\rho_d^r$ | [m$^2$s$^{-2}$] | normalized pressure perturbation |
| $\boldsymbol{k}$ | [1] | vertical unit vector |
| $B$ | [m s$^{-2}$] | buoyancy |
| $F_X$ | [(unit of $X$) s$^{-1}$] | forcing of $X$ (surface fluxes, radiation, absorbers, subsidence, ...) |
| $X^e, X^r$ | [(unit of $X$)] | environmental/reference profile of $X$ |
| $E_c, E_e, C = E_c - E_e$ | [s$^{-1}$] | condensation/evaporation rate and their balance |
| $g$ | [m s$^{-2}$] | magnitude of Earth's gravitational acceleration |
| $\epsilon = R_v/R_d - 1$ | [1] | |
| $\rho, \rho_d$ | [kg m$^{-3}$] | density of air/dry air |
| $S = \mathrm{d}_z\theta_v/\theta_v$ | [m$^{-1}$] | non-dimensional stability of the atmosphere |
| $r, r_d$ | [m] | wet/dry radius of a SD |
| $\kappa$ | [1] | hygroscopicity parameter of a SD |
| $\xi$ | [1] | multiplicity of a SD |
| $e_s$ | [Pa] | saturation partial pressure of vapor |
| $N_c$ | [m$^{-3}$] | concentration of cloud droplets in cloudy grid cells |
| $N_{\mathrm{SD}}$ | [1] | initial number of SDs per grid cell |
| $\Delta t$ | [s] | time step length of the dynamical core |
| $z_i$ | [m] | mean height of the $q_t = 8$ g kg$^{-1}$ isosurface |
| $w_t, w_{\mathrm{LS}}$ | [m s$^{-1}$] | terminal velocity of a SD/large-scale subsidence velocity |
| $\rho_w$ | [kg m$^{-3}$] | density of water |
| $\mathcal{D}_X$ | [(unit of $X$) s$^{-1}$] | SGS model forcing of $X$ |
| $K_m, K_h, K_q$ | [m$^2$ s$^{-1}$] | eddy viscosity/eddy diffusivities |
| $\boldsymbol{E}$ | [s$^{-1}$] | deformation tensor |
| $c_s$ | [1] | Smagorinsky constant |
| $\lambda$ | [m] | mixing length |
| $\Delta$ | [m] | cell length scale |
| $c_L$ | [1] | parameter characterizing mixing length growth rate near the ground |
| Pr, Ri | [1] | Prandtl/Richardson number |

## Appendix B: Condensation substepping algorithm

Consider condensation of SDs within cell $i$ at time step $n$. Vector of thermodynamic conditions in that cell at the moment right before condensation is calculated is denoted by $\boldsymbol{\psi}_i^{[n]} = \left(\theta^{[n]}, q_v^{[n]}\right)_i$. Number of time steps is denoted by $S_c$ and substeps are indexed by $\nu$, starting at $\nu = 1$. Super-droplets within cell $i$ are numbered by $\mu$. Vector of thermodynamic conditions that a given SD experiences at substep $\nu$ is denoted by $\breve{\boldsymbol{\psi}}_\mu^{[\nu]}$. Using this notation, the substepping algorithm is

$$\breve{\boldsymbol{\psi}}_\mu^{[\nu+1/2]} = \breve{\boldsymbol{\psi}}_\mu^{[\nu]} + \frac{\boldsymbol{\psi}_i^{[n]} - \breve{\boldsymbol{\psi}}_\mu^{[\nu=1]}}{S_c}, \tag{B1}$$

$$r_\mu^{2[\nu+1]} = r_\mu^{2[\nu]} + \frac{\Delta t}{S_c} \left. \frac{dr^2}{dt} \right|_{r_\mu^{2[\nu+1]}, \breve{\boldsymbol{\psi}}_\mu^{[\nu+1/2]}}, \tag{B2}$$

$$\breve{\boldsymbol{\psi}}_\mu^{[\nu+1]} = \breve{\boldsymbol{\psi}}_\mu^{[\nu+1/2]} + \boldsymbol{A} \frac{4}{3} \frac{\pi \rho_w V}{\rho_d^r} \sum_{\mu=1}^{\mu=N_i^{[n]}} \xi_\mu \left[ \left(r_\mu^{2[\nu+1]}\right)^{3/2} - \left(r_\mu^{2[\nu]}\right)^{3/2} \right], \tag{B3}$$

where $r_\mu^2$ is the square of the wet radius of the $\mu$-th SD, $N_i^{[n]}$ is the number of SDs in cell $i$ at time step $n$ and $\boldsymbol{A} = (\theta^e l_v / (c_{pd} T^e), -1)$. Sum in eq. (B3) is done over all SDs in cell $i$ at time step $n$. For details of the predictor-corrector algorithm for calculation of the change of radius in eq. (B2), see Eqs. (17)-(19) in Arabas et al. (2015). After the last substep, the value of $\breve{\boldsymbol{\psi}}_\mu^{[\nu=S_c]}$ is the same for all SDs in the cell and the condensational RHS returned from the condensation algorithm is

$$R_c^{[n]} = \frac{\breve{\boldsymbol{\psi}}_{\mu=1}^{[\nu=S_c]} - \boldsymbol{\psi}_i^{[n]}}{\Delta t}. \tag{B4}$$

The initial value $\breve{\boldsymbol{\psi}}_\mu^{[\nu=1]}$ is equal to the thermodynamic conditions after condensation finished in the previous time step. Two ways of defining $\breve{\boldsymbol{\psi}}_\mu^{[\nu=1]}$ are considered that differ in the spatial cell from which this initial condition is diagnosed:

$$\breve{\boldsymbol{\psi}}_\mu^{[\nu=1]} = \left(\boldsymbol{\psi}^{[n-1]} + R_c^{[n-1]}\right)_{i(n-1)}, \tag{B5}$$

referred to as *per-particle* substepping, and

$$\breve{\boldsymbol{\psi}}_\mu^{[\nu=1]} = \left(\boldsymbol{\psi}^{[n-1]} + R_c^{[n-1]}\right)_{i(n)}, \tag{B6}$$

a procedure we call *per-cell* substepping. The notation $i(n)$ stands for the index of the cell in which the $\mu$-th SD was at time step $n$. The *per-cell* substepping is less accurate, but requires less computational time and uses less memory. The reason is that in the *per-cell* method, all SDs in a given cell have the same values of $\breve{\boldsymbol{\psi}}_\mu^{[\nu]}$. Moreover, values of pressure and density do not need to be substepped in the *per-cell* method, since they are constant in time in each cell.

We expect the difference between the two substepping algorithms to be particularly large near a moving cloud edge. We test this hypothesis by simulating cloud edge advection in an idealized one-dimensional setup. Consider two spatial cells, one at saturation ($q_l = 0.029$g/kg, $N_c \approx 52$cm$^{-3}$) and the other subsaturated (relative humidity of 94%). The boundaries are periodic and the only processes are diffusional growth and advection. A single time step ($\Delta t = 2$s) is performed, in which Eulerian fields

and SDs are advected with the Courant number equal to 1. The expected result is that the two cells exchange their contents without any condensation/evaporation taking place. The simulations are conducted for different substepping algorithms and different numbers of substeps. Deviations from the expected result are presented in table B1.

**Table B1.** Errors caused by substepping in the one-dimensional simulation of cloud edge advection. The error is defined as $\epsilon = \left(q_l^{\text{sim}} - q_l^{\text{exp}}\right) / q_l^{\text{exp}}$, where $q_l$ is the liquid water mixing ratio diagnosed at the end of the simulation from the cell to which cloud droplets were advected, i.e. the cell that initially was subsaturated. The superscripts "sim" and "exp" denote the simulated and expected values, respectively.

| substepping algorithm | number of substeps | $\epsilon$ [%] |
|---|---|---|
| *per-cell* | 1 | 0 |
| | 2 | -26 |
| | 5 | -40 |
| | 10 | -44 |
| *per-particle* | 1 | 0 |
| | 2 | 0 |
| | 5 | 0 |
| | 10 | 0 |

The *per-cell* algorithm causes artificial evaporation of droplets and this error increases with the number of substeps, while the *per-particle* algorithm gives correct results independently of the number of substeps. Therefore we conclude that in simulations in which there is a lot of cloud edge advection it is necessary to use the *per-particle* algorithm.

However, in simulations with little cloud edge advection, the per-cell algorithm might be sufficient. We check this by using different substepping algorithms in a two-dimensional marine startocumulus simulation (the DYCOMS RF02 case, cf. section 4). To limit differences in dynamics between runs, we use the piggybacking approach (Grabowski, 2014): velocity fields from a dynamical "driver" simulation with $\Delta t = 0.1$s are used in two other "piggybacking" simulations with $\Delta t = 1$ s. No substepping is done in the driver simulation and the piggybacking simulations have 10 substeps for condensation, one with the *per-cell* and the other with the *per-particle* algorithm. Timestep length in the "driver" run is shorter than in the piggybacking runs, because we want to properly model condensation in the "driver" simulation without substepping, in order to obtain reference results. Obviously, due to the difference in time step length, velocity field used in "piggybacking" simulations is not exactly the same as in the "driver" simulations. However, averaged vertical profiles of moments of the vertical velocity are similar in the "driver" and "piggybacker", so we expect concentration of cloud droplets in the "piggybacker" to be similar to that of the "driver". We are interested in cloud droplet concentration, because droplet activation is the process that requires short time steps for condensation. To limit variability between runs caused by the Monte Carlo scheme used to initialize SD radii, a large number of SDs is used in these simulations ($N_{SD} = 1000$). Vertical profiles of concentration of cloud droplets

from the "driver" and "piggybacker" runs are shown in fig. B1. Interestingly, the *per-cell* algorithm is in better agreement with the "driver" simulation than the *per-particle* algorithm. The latter slightly underestimates concentration of cloud droplets, by ca. 5%. Nevertheless, we conclude that both substepping algorithms work well in simulations in which cloud edge does not move significantly.

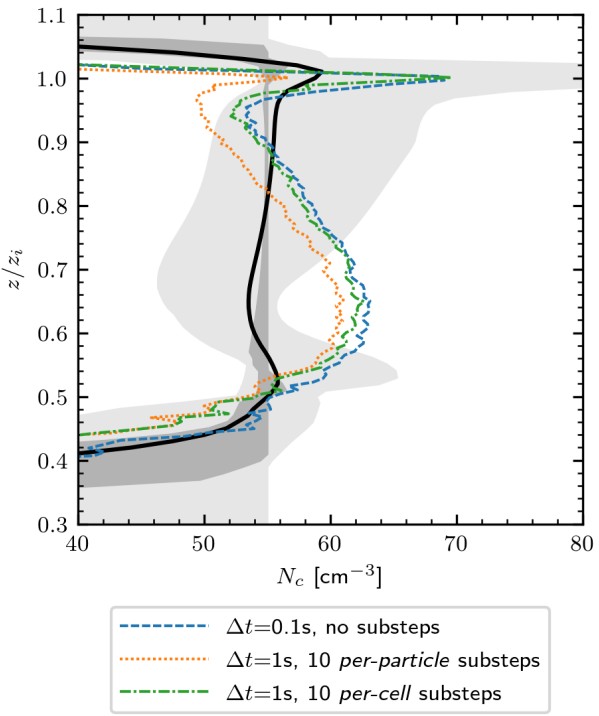

**Figure B1.** Vertical profiles of concentration of cloud droplets from the 2D piggybacking simulations with different substepping algorithms, averaged over the 2h-4h period. The black line and shaded regions show results from Ackerman et al. (2009) (cf. fig. 3).

## 5 Appendix C: Software implementation

UWLCM is coded in the C++ language. It relies heavily on two C++ libraries developed by the cloud modeling group at the University of Warsaw: libmpdata++ (Jaruga et al., 2015) for the Eulerian component and libcloudph++ (Arabas et al., 2015) for the Lagrangian component of the model. Structure of the libmpdata++ and libcloudph++ codes and how they are used in UWLCM is schematically depicted in Fig. C1.

10      libmpdata++ is a set of solvers for the generalized transport equations that use the MPDATA advection scheme. The solvers are organized in a hierarchy, ordered from solvers for simple flows to solvers for more complex flows. Each more complex solver inherits from the simpler solver in the hierarchy. Such design simplifies code development, maintenance and reusability.

Apart from the hierarchy of solvers, libmpdata++ contains three other independent modules: boundary conditions, concurrency handlers and output handlers, all of which are used in UWLCM.

libcloudph++ is an implementation of three microphysical models: SDM, a single-moment bulk model and a double-moment bulk model. The SDM is implemented using the Thrust library and the CUDA programming language. Thanks to that, the SDM can be run on multi-threaded CPUs as well as on multiple GPUs.

UWLCM code is built on top of the libmpdata++ solvers. Separate parts of the UWLCM code handle different types of simulations. The "piggybacking" code makes it possible to run kinematic simulations, i.e. simulations with a prescribed velocity field. The "2D/3D" part of the code handles the dimensionality of the problem. The "forcings" code specifies external forcings, so it is the part of the code that depends on simulation setup. The "microphysics" module is responsible for handling the choice of microphysics (only Lagrangian microphysics is available in the current UWLCM release). Thanks to such code structure, different types of simulations, e.g. 2D and 3D simulations, different simulation setups or kinematic simulations, are using mostly the same source code. The highest performance is achieved when UWLCM is run on a system with GPUs. In that case, the Eulerian component is calculated on CPUs and the Lagrangian component on GPUs. Large part of these computations is done simultaneously (cf. fig. 1). The UWLCM code is open-source, under a version-control system and available from a git repository. Model output is done in the HDF5 format, ready for plotting in *Paraview*. UWLCM also includes simple software for plotting time series and vertical profiles. A number of test programs was developed for UWLCM, libcloudph++ and libmpdata++. UWLCM currently can run in parallel only on shared-memory systems. An implementation for distributed-memory systems is currently under development. UWLCM code is inspired by the *icicle* kinematic model developed by S. Arabas and A. Jaruga (https://github.com/igfuw/libcloudphxx/models/kinematic_2D).

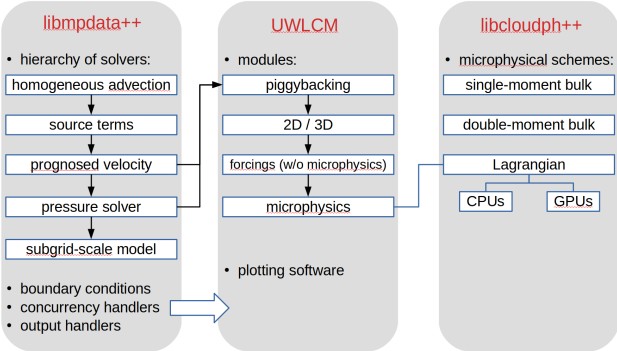

**Figure C1.** Schematic depiction of the structure of the code of UWLCM, libmpdata++ and libcloudph++. Black arrows denote inheritance between classes.

*Author contributions.* PD developed the model code with contributions from MW. PD performed the simulations. PD and HP prepared the manuscript with contributions from MW.

*Acknowledgements.* We thank Wojciech Grabowski for discussions about the model formulation and about the manuscript. We are grateful to the AGH Cyfronet computation center for providing computing power. This research was supported by the Polish National Science Center grant 2016/23/B/ST10/00690.

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
