# Peer review of "University of Warsaw Lagrangian Cloud Model (UWLCM) 1.0: a modern Large-Eddy Simulation tool for warm cloud modeling with Lagrangian microphysics"

_Geoscientific Model Development, 2018_

## Referee Comment (RC1) · Anonymous Referee #1 · 21 Feb 2019

**Review of "University of Warsaw Lagrangian Cloud Model (UWLCM) 1.0: a modern Large-Eddy Simulation tool for warm cloud modeling with Lagrangian microphysics" by Dziekan et al. (doi:10.5194/gmd-2018-281)**

The manuscript describes the University of Warsaw Lagrangian Cloud Model (UWLCM), a combination of Eulerian LES and Lagrangian cloud microphysics. Therefore, the manuscript can be seen as a continuation of Arabas et al. (2015, doi:10.5194/gmd-8-1677-2015) and Jaruga and Pawlowska (2018, doi:10.5194/gmd-2018-96), which describe earlier versions of the Lagrangian microphysics now applied in UWLCM. Additionally, the manuscript addresses different approaches to time sub-stepping and the number of simulated Lagrangian particles necessary for convergence, which are not only of value for UWLCM but also the larger Lagrangian cloud modeling community.

The manuscript fits the scope of Geoscientific Model Development. It is generally well written, but demands some language editing. Overall, there are a few slightly major comments and quite some minor issues that need to be addressed before advancing in the publication process.

**Major Comments**

*Diffusion of superdroplets* (p. 10, ll. 5 – 6; p. 5, l. 14; p. 13, l. 25). The motion of superdroplets in only determined by the resolved-scale LES air motion. Turbulent diffusion, which is considered in the LES implicitly due to numerical diffusion, is not considered for the superdroplets. This underestimates the diffusion of superdroplets and liquid water in all simulations, indicating that the fields of water vapor, temperature, and liquid water are not in physical agreement. I appreciate that the authors are candid about this issue, but they should address the implications of this discrepancy more clearly. Especially because there are methods available and to consider subgrid-scale motion of Lagrangian particles (e.g., Weil et al. 2004, doi:10.1175/JAS-3302.1), which are already in use in other Lagrangian cloud models (Sölch and Kärcher 2010, doi: 10.1002/qj.689; Hoffmann et al. 2017, doi: 10.1175/JAS-D-16-0220.1). One example where this neglect probably matters is the number of simulated cloud droplets $N_C$. The authors state that $N_C$ is higher in UWLCM compared to other models (p. 13, l. 25). They explain this by the lack of numerical diffusion. This is right. However, the neglected turbulent diffusion of superdroplets also contributes to a higher $N_C$ and needs to be mentioned.

*Comparison of different time sub-stepping schemes.* The comparison of the per-particle and per-cell sub-stepping approaches with a simulation without sub-stepping but a commensurately reduced timestep of 0.1 s is not very helpful due to the strong interaction of microphysics and dynamics. This becomes very clear for the three-dimensional simulations, in which the 0.1 s simulation enables a more detailed, and probably more adequate representation of this interaction. As a result, the entrainment rates vary significantly among the different model setups as seen in Fig. 5b, with commensurate effects on the liquid water path (decreases due to stronger entrainment), cloud base height (increases due to stronger entrainment), and indirectly precipitation (increases with liquid water path). To derive useful conclusions, it is necessary to untangle dynamical and microphysical effects. Therefore, I strongly suggest using either a kinematic driver providing each setup the identical dynamical forcing or to use the piggy-backing approach, which is actually part of UWLCM as stated on p. 22, ll. 12 – 13.

**Minor Comments**

P. 2, l. 9: Please clarify: Automated tests for what?

P. 2, l. 16 – 17: Focusing on precipitation is one aspect. Cloud cover might be an additional and very important second aspect to consider since precipitation might result in the transition from closed to open cells.

P. 3, l. 6 – 8: How does the auxiliary environmental state increase the *precision* of numerical calculations? Usually, these environmental states are necessary requirements to solve the system of

equations. Furthermore, the word precision usually refers to the number of significant digits of the solution. I do not believe that this is meant by authors.

P. 3, Eq. (3): It is explained later, but a brief description of what $\pi$ is might be helpful at this point.

P. 4, Eq. (10): What are r and $r_d$?

P. 4, l. 12: What is so special about this definition of the relative humidity (the ratio of actual and saturation water vapor mixing ratio) to cite Lipps and Hamler (1982)? E.g., Clark (1973, doi:10.1175/1520-0469(1973)030<0857:NMOTDA>2.0.CO;2) defined the supersaturation (his Eq. (15)) also as the ratio between actual and saturation water vapor mixing ratio.

P. 4, l. 13: Consider replacing 0.622 by the ratio of the specific gas constant of dry air to the specific gas constant of water vapor (i.e., $R_a/R_v$).

P. 4, l. 14: Please comment if D and K include gas kinetic or ventilation effects.

P. 4, l. 19: For clarity, add "real" between "two" and "droplets".

P. 4, ll. 24 – 25: Superdroplets do not collide. Equation (12) states the probability that one real droplet of superdroplet j (or k) collects any real droplet of superdroplet k (or j).

P. 4, l. 27: Starting from (12), there are not necessarily $\xi_j$ pairs of real droplets coalescing. The correct number is min $(\xi_j, \xi_k)$.

P. 5, l. 14: The sedimentation velocity is explicitly considered in the motion of superdroplets. I believe this counts the (admittedly small) contribution of sedimentation twice since it is already considered in the LES velocity vector **u**, according to (3).

P. 5, ll. 23 – 24: Equation (14) is still "Eulerian" in the sense that is contains an advection term.

P. 6, ll. 3 – 5: State clearly that $\pi$ is the pressure perturbation. Furthermore, I think the introduction "[that] it is characteristic for anelastic models that the pressure perturbation does not follow the ideal gas law" causes more confusion than clarification. I would omit it.

P. 6, 13 – 14, Shima et al. (2009) were not the first to advocate the integration of the squared wet radius. See, e.g., J.-P. Chen (1992): *Numerical simulations of the redistribution of atmospheric trace chemicals through cloud processes* (Doctoral dissertation, Pennsylvania State University), especially his Eq. (3.81).

P. 6, ll. 18 – 19: In what sense is condensation a fast process here? I think you need to be more specific. Arnason and Brown (1971, doi:10.1175/1520-0469(1971)028<0072:GOCDBC>2.0.CO;2) showed that for condensation a timestep corresponding to the phase relaxation timescale is sufficient, i.e., about 1 s or even longer for clean clouds. The requirement for a 0.1 s timestep arises, in my eyes, from the rapid change in droplet radius during growth at small radii. This is a well-known feature of stiff differential equations, as it is the case for the diffusional growth equation for droplets. Furthermore, how do you know that a sub-stepping timestep of 0.1 s is sufficient? In similar simulations of Grabowski et al. (2011, doi:10.1016/j.atmosres.2010.10.020) an initial timestep of $10^{-6}$ s that might increase to 0.1 s is used to integrate the diffusional growth equation (see their Appendix). Of course, they integrated the linear growth equation (dr/dt) and not the quadratic ($dr^2/dt$) as done here. But additional stand-alone integrations of superdroplets with different aerosol masses and a prescribed supersaturation using different timestep lengths are necessary to verify if a 0.1 s sub-timestep is actually sufficient.

P. 8, l. 8: Consider changing "a pair" to "the same pair" for clarity.

P. 8, l. 26: RHS of what?

P. 8, ll. 30 – 31: These sentences contradict each other since the UWLCM contains an LCM and an LES. Therefore, specify "[all] of the model dependent variables" more precisely.

P. 9, ll. 9 – 10, Fig. 2: Figure 2 confuses me. If only the shaded part is used as a coalescence cell, certain volumes filled with superdroplets are neglected in the collection process. However, I do not believe that this is what the authors are doing. Could it be the case that the lowest line of grid point always equals the first, and that the right-most column of grid point equals the left-most? In other words, how do the authors implement so-called ghost layers of grid points to facilitate a cyclic model domain?

P. 9, l. 18 – 19: Important for the formation of drizzle is the is the microphysical model, and usually not the LES dynamical core.

P. 10, ll. 25 – 26: Please comment on these options if they are essential for the conducted simulations. If they are not essential, I would omit this sentence for clarity.

P. 10, ll. 30 – 32: State that turbulence in two dimensions behaves fundamentally different from turbulence in three dimensions.

P. 10, l. 33 – p. 11, l. 1: Small random perturbations are not the reason for the variability, it is a fundamental property of a chaotic system, reacting to small changes in the initial values.

Figs. 3 – 6: For the final version of this manuscript, please make sure that the location of the figures matches the text.

P. 12, ll. 1 – 2: The entrainment is usually not calculated from the increase of the inversion height alone. Commonly, the subsidence velocity at cloud top height is subtracted.

P. 12, ll. 23 – 24: I suggest rewriting this sentence to: "[…] where the autoconversion efficiency increases with $N_{SD}$."

P. 12, ll. 26 – 28: Since the characteristics of turbulence in two dimensions are fundamentally different from three dimensions, the better agreement with observations must be seen as purely coincidental.

P. 13, l. 29: Why is the iLES approach responsible for the simulated behavior of the third moment of the vertical velocity?

P. 13, l. 33 – p. 15, l. 2: Spurious cloud edge supersaturations are known to result in the artificial activation of cloud droplets at the top of stratocumulus (e.g., Stevens et al. 1996, doi:10.1175/ 1520-0493(1996)124,1034:TSPOCE.2.0.CO;2). Physical activations are largely impossible there since the top of stratocumulus is not dominated by strong, long-lasting updrafts resulting in physical supersaturations.

P. 17, ll. 1 – 2: Maybe it is worthwhile to add references to the models DHARMA and RAMS.

P. 17, sec. 4.5: How is activation determined? I assume a droplet is considered activated when it exceeds a critical radius. This is a valid assumption if the aerosol is small, and diffusional growth is not kinetically limited. However, for aerosols smaller than 0.1 μm, the typical timescale for activation is usually similar or even smaller than the timestep of the applied model, making the treatment of

activation in UWLCM, DHARMA, and RAMS practically identical. The activation timescale becomes only important if the aerosol is large, typically larger than 0.1 μm in radius, for which the critical radius exceeds a couple of micrometers. However, once located in a saturated environment, these *inactivated* particles exhibit behavior very similar to regularly activated droplets once their wet radius exceeds one micrometer, beyond which curvature and solute effects are usually negligible. Accordingly, the reduced susceptibility of aerosol activation on the cloud-base supersaturation maximum might also be just a result of the applied criterion for activation, which is not appropriate for the entire aerosol spectrum.

P. 17, ll.. 26 – 27: Please clarify: The cloud-base supersaturation maximum still causes activation in UWLCM, but it might not have an as immediate effect as in other cloud models because of the (presumably) applied criterion for activation (see last comment).

P. 19, ll. 10 – 11: I agree, that the number of superdroplets has no impact on domain-averaged quantities. However, it might be worthwhile to refer to the study of Schwenkel et al. (2018, doi:10.5194/gmd-11-3929-2018) in which small-scale effects of the superdroplets concentration are addressed.

**Technical Comments**
P. 5, l. 4: Change format of citation: "[…] in Gillespie (1972), […]", not  "[…] in (Gillespie, 1972), […]"

P. 10, l. 16: Change format of citation: "[…] in Ackerman et al. (2009), […]", not  "[…] in (Ackerman et al., 2009), […]"

---

## Referee Comment (RC2) · Anonymous Referee #2 · 6 Mar 2019

Review of Dziekan et al, GMDD, 2019

**Summary**

This manuscript introduces a LES model with a dynamical core model and a fully coupled Lagrangian microphysical model for warm clouds. Both model parts have been described previously in two separate GMD publications. The manuscript repeats the description of the main components of the dynamical core and the microphysics part in a condensed form. A case study of a drizzling marine stratocumulus, which has been used in a model benchmark study (11 participating LES models in a 2009 publication), has been performed and the simulation results have been compared to the 11 LES models.

The content of the present manuscript is well suited to be published in GMD and I recommend publication after major revisions. I believe that the manuscript would benefit from inclusion of a second test case.

**Major comments**

1. As the main components have been described elsewhere, and the coupling of the two parts seems to be rather straightforward, a stronger emphasis could be put on the model verification. You compare it to an ensemble of 11 reference models with quite some spread. But no one knows what the truth is. So far, I am not really sure what conclusions are to be drawn from your comparison and how I should interpret your results? Can you conclude anything e.g. from the fact that your model lies above or below the ensemble mean for some physical quantity? Better describe what you expect from your comparison exercise. Focusing on one specific test case gives only a snapshot of the model's overall behaviour and it is not clear how robust and general your findings are. It would be interesting to see how your model behaves in another well-chosen test case.

2. To be frank, resorting to the iLES approach comes in handy as you don't have to implement a SGS scheme. I could live with it if your model is purely Eulerian. As the Lagrangian model has no implicit numerical diffusivity (neither in spectral nor spatial space) and the iLES approach is not applicable in the microphysics part, SGS random perturbation velocities could be included in the transport equation of the superdroplets in order to mimic subgrid scale motions. However, without a proper SGS scheme that estimates TKE it is not straightforward to prescribe such perturbations. This shortcoming should be clearer mentioned.

**Minor comments**

- P1. Last row: Isn't libmpdata++ the dynamical core? What does it mean "it is built on top of" it?
- p.4, l.22: Without defining what a collision between two SDs is, it makes no sense to say the probability needs to be increased. Please rephrase.
- p.5, l.14: I do not understand the inclusion of $w_{LS}$. This would mean that the SDs move relative to the surrounding (Eulerian) air!?

- Sec 3.2.: The implementation of the various condensation algorithms is not clear to me. Given that $\Psi_{new}$ and $\Psi_{old}$ are known, you do a linear time interpolation between the two values. And the difference between the two approaches is the choice of the grid box from which you pick the $\Psi$ values. What I stumble upon is the quantity $\Psi_{new}$. Is it known beforehand? In my understanding, sub-stepping would simply mean that condensation (growth of droplets, depletion of water vapour and latent heat release) is treated with a smaller time step and clearly involves a dynamic update of the variables $\theta$ and $q_v$ in each sub-time step.
- Sec 3.3: I do not fully understand why you solve a prognostic equation for $q_l$ in the Eulerian model part. Wouldn't it suffice to diagnose $q_l$ from the SDs? I understand that $q_l$ is used for the computation of the buoyancy term (Eq. 6). Do you need it elsewhere? Can you estimate the error of using two different definitions of $q_l$? You write that you want to avoid an additional synchronization? Would this issue still matter in a parallelised implementation?
- Sec 4.1: Can you comment why you use a split definition (Hall +Davies) for the collision efficiencies?
- Sec 4.2: In particular, the differences between the *per-cell* and *per-particle* approach are so small that I am not fully convinced that the one is superior over the other one. It would also help to see the spread of the 10-member ensemble of a specific 2D simulation. Is it really significant that in the one case the $N_c$-profiles slightly decrease with altitude, whereas in the other case they slightly increase? Can you be sure that in other test case, your finding (superiority of the *per-particle*) would be the same?
This is one example why I recommend a second test case.

**Typos, language issues and other formal things:**
- In general, the usage of articles "a" and "the" is not correct on several occasions. Sometimes you miss the article, sometimes it is misplaced. Please try your best, the rest will be handled by Copernicus services.
- The Exner function pi should be defined close to Eq. 3
- p.4, l.16: collisionS
- There is a difference between which and that: https://www.wisegeek.com/what-is-the-difference-between-that-and-which.htm
Accordingly, "which" in p.4, l.19 and l.24 must be replaced by "that". There might be more such mistakes.
- p.4, l. 23: dropletS
- p.12, l.10. not sure if "VAR" is self-explaining?
- p.12, l.31: visible IN
- p.13, l.11: impact IN 3D simulations than IN 2D simulations
- Caption of Fig. 4: Please correct "On the vertical axis is height …"
- P.17, l.17: concentration

---

## Author Comment (AC1) · 13 May 2019

**Response to the reviews of "University of Warsaw Lagrangian Cloud Model (UWLCM) 1.0: a modern Large-Eddy Simulation tool for warm cloud modeling with Lagrangian microphysics" by Dziekan et al. (doi:10.5194/gmd-2018-281)**

May 13, 2019

We thank the reviewers for the work they have put into improving the manuscript. Before we respond to their comments, we need to point out that an error was found in our implementation of the radiation scheme used in the DYCOMS RF02 simulations. The error resulted in wrong distribution of radiative flux within cloud layer - radiative cooling was decreasing temperature practically only in the uppermost cloudy cell and only the lowermost cloudy cell was being radiatively heated. The error has been fixed and all simulations were repeated. The most profound difference in results is that the LWP has become higher and that there is more surface precipitation in 2D simulations. The large amount of surface precipitation in 2D simulations prompted us to study how precipitation formation in Lagrangian microphysics depends on the time step for coalescence. This sensitivity study is now presented in the section about 2D simulations.

Answer to the Anonymous Referee #1.

**Major Comments**

**Diffusion of superdroplets (p. 10, ll. 5 6; p. 5, l. 14; p. 13, l. 25).** The motion of superdroplets in only determined by the resolved-scale LES air motion. Turbulent diffusion, which is considered in the LES implicitly due to numerical diffusion, is not considered for the superdroplets. This underestimates the diffusion of superdroplets and liquid water in all simulations, indicating that the fields of water vapor,

temperature, and liquid water are not in physical agreement. I appreciate that the authors are candid about this issue, but they should address the implications of this discrepancy more clearly. Especially because there are methods available and to consider subgrid-scale motion of Lagrangian particles (e.g., Weil et al. 2004, doi:10.1175/JAS-3302.1), which are already in use in other Lagrangian cloud models (Slch and Krcher 2010, doi: 10.1002/qj.689; Hoffmann et al. 2017, doi: 10.1175/JAS- D-16-0220.1). One example where this neglect probably matters is the number of simulated cloud droplets N C . The authors state that N C is higher in UWLCM compared to other models (p. 13, l. 25). They explain this by the lack of numerical diffusion. This is right. However, the neglected turbulent diffusion of superdroplets also contributes to a higher N C and needs to be mentioned.

The issue of SGS diffusion was also brought up by the Reviewer #2. To resolve it, we have added results of 3D simulations using the Smagorinsky scheme, with and without SGS motion of Lagrangian particles. The section presenting 3D simulations is now focused on comparing these different SGS modeling techniques. After fixing the radiative scheme, ILES gives larger LWP than reference simulations. It is shown that to obtain agreement with the reference simulations it is necessary to use the Smagorinsky scheme and to include the SGS motion of Lagrangian particles.

**Comparison of different time sub-stepping schemes. The comparison of the per-particle and per-cell sub-stepping approaches with a simulation without sub-stepping but a commensurately reduced timestep of 0.1 s is not very helpful due to the strong interaction of microphysics and dynamics. This becomes very clear for the three-dimensional simulations, in which the 0.1 s simulation enables a more detailed, and probably more adequate representation of this interaction. As a result, the entrainment rates vary significantly among the different model setups as seen in Fig. 5b, with commensurate effects on the liquid water path (decreases due to stronger entrainment), cloud base height (increases due to stronger entrainment), and indirectly precipitation (increases with liquid water path). To derive useful conclusions, it is necessary to untangle dynamical and microphysical effects. Therefore, I strongly suggest using either a kinematic driver providing each setup the identical dynamical forcing or to use the piggy-backing approach, which is actually part of UWLCM as stated on p. 22, ll. 12 13.**

Following the comment, we performed 2D piggybacking simulations in which flow field from a simulation with $\Delta t = 0.1$ s is used to drive two simulations with $\Delta t = 1$ s and different substepping techniques. The conclusion is that, for stratocumulus clouds, the *per-cell* algorithm is better, but the *per-particle*

algorithm also works well. All the stratocumulus simulations presented in the revised paper use *per-cell* substepping. We expect *per-cell* substepping to give errors for a fast moving cloud edge. To test this, we also present idealized 1D simulations of a moving cloud edge. There, *per-cell* substepping causes significant errors and *per-particle* algorithm works well. Discussion of differences between results of different substepping algorithms has been moved to the Appendix B. Section of the main text discussing 2D simulations deals now with sensitivity of Lagrangian microphysics to the coalescence time step and to the number of computational particles.

**Minor Comments**

**P. 2, l. 9: Please clarify: Automated tests for what?**

Some more information has been added:
" A set of automated tests greatly helps in ensuring the correctness of the model. The automated tests include a 2D moist thermal simulation, a 2D kinematic stratocumulus simulation and a test of different combinations of model options. Moreover, modeling of physical processes, e.g. condensation, advection, coalescence, sedimentation, is tested separately by the libmpdata++ and libcloudph++ test suites. "

**P. 2, l. 16 17: Focusing on precipitation is one aspect. Cloud cover might be an additional and very important second aspect to consider since precipitation might result in the transition from closed to open cells.**

The paper introduces a new model, therefore we focus on basic cloud properties and do not study more complex behavior. However, in the discussion of 3D results we now mention that cloud cover is close to 100% in our simulations:
" Also, cloud cover, defined as fraction of columns with LWP $> 20 \mathrm{gm}^{-2}$, is close to 100 % in all 3D UWLCM simulations. "

**P. 3, l. 6 8: How does the auxiliary environmental state increase the precision of numerical calculations? Usually, these environmental states are necessary requirements to solve the system of equations. Furthermore, the word precision usually refers to the number of significant digits of the solution. I do not believe that this is meant by authors.**

Our notion of the environmental state (also known as the ambient state) is distinct from the reference state. Introduction of environmental states is optional. However, their able choice can facilitate the design of initial or boundary conditions, improve the conditioning of the elliptic boundary value problems,

and/or enhance the accuracy of calculations in finite-precision arithmetics (Smolarkiewicz et al., 2014; **?**). We admit that the word "precision" was confusing and we have changed it to "accuracy".

**P. 3, Eq. (3): It is explained later, but a brief description of what is might be helpful at this point.**

The sentence right after Eq. (3) now states: "where $D_t$ denotes the material derivative: $D_t = \partial_t + \vec{u} \cdot \nabla$ and $\pi$ is normalized pressure perturbation."

**P. 4, Eq. (10): What are $r$ and $r_d$ ?**

They are the dry and wet radius, respectively. We believe this should be clear, as it is stated in the first paragraph of the section and in the table in Appendix A.

**P. 4, l. 12: What is so special about this definition of the relative humidity (the ratio of actual and saturation water vapor mixing ratio) to cite Lipps and Hamler (1982)? E.g., Clark (1973, doi:10.1175/1520-0469(1973)030¡0857:NMOTDA¿2.0.CO;2) defined the supersaturation (his Eq. (15)) also as the ratio between actual and saturation water vapor mixing ratio.**

Small differences in definition of relative humidity can have visible impact on results. RH $= q_v/q_{vs}$ is an approximation of the more correct RH $= e/e_s$. More importantly, it is not obvious for us how to calculate dry air partial pressure $p_d$ in the anelastic approximation. Should it follow from the ideal gas law, like vapor partial pressure does? Or should it be selected so that $e + p_d = p^e$, where $e$ is calculated from the ideal gas law? Lipps and Hemler use the second approach and we also adopt it to be consistent with the Lipps-Hemler approximation, so we explicitly reference their paper.

**P. 4, l. 13: Consider replacing 0.622 by the ratio of the specific gas constant of dry air to the specific gas constant of water vapor (i.e., R a /R v ).**

Done.

**P. 4, l. 14: Please comment if D and K include gas kinetic or ventilation effects.**

They include both, an appropriate comment has been made in the text.

**P. 4, l. 19: For clarity, add real between two and droplets.**

Done.

**P. 4, ll. 24 25: Superdroplets do not collide. Equation (12) states the probability that one real droplet of superdroplet j (or k) collects any real droplet of superdroplet k (or j).**

Incorrectly, we were using the words "collide" and "coalesce" to describe coalescence. This has been fixed by changing instances of "collide" with "coalesce". The nomenclature that superdroplets coalesce is used following Shima et al. (2009). How we interpret coalescence of superdroplets is explained in the paragraph directly following eq. (12) (eq. 17 in the revised manuscript). Equation (12) does not state the probability that one real droplet of superdroplet j (or k) collects any real droplet of superdroplet k (or j). Instead, it states the probability that each of $\xi_j$ real droplets of superdroplet j coalesces with a single real droplet of superdroplet k, where j and k labels are chosen so that $\xi_j \leq \xi_k$.

**P. 4, l. 27: Starting from (12), there are not necessarily $\xi_j$ pairs of real droplets coalescing. The correct number is min $(\xi_j, \xi_k$ ).**

As stated on p.4 l. 28, SDs are labeled so that $\xi_j \leq \xi_k$. Then min $(\xi_j, \xi_k$ ) $= \xi_j$. To make this convention more clear, now we introduce it right after eq. (12):
" where SDs are labeled so that $\xi_j \leq \xi_k$ and this convention is assumed throughout the rest of this paragraph. "

**P. 5, l. 14: The sedimentation velocity is explicitly considered in the motion of superdroplets. I believe this counts the (admittedly small) contribution of sedimentation twice since it is already considered in the LES velocity vector u, according to (3).**

We believe that the Referee has the large scale subsidence in mind and not sedimentation of droplets. Large scale subsidence is added as an RHS of the prognostic Eulerian variables. Adding it to the RHS of $\vec{u}$ in eq. (3) means that the velocity vector is moved downwards by large scale subsidence, but does not mean that the vertical velocity component includes the large scale subsidence velocity. Therefore this velocity is added to velocities of superdroplets and that way it is included only once as it should be.

**P. 5, ll. 23 24: Equation (14) is still Eulerian in the sense that is contains an advection term.**

The equation is now written in a form that is usually referred to as Lagrangian: $D_t \psi = R$.

**P. 6, ll. 3 5: State clearly that $\pi$ is the pressure perturbation. Furthermore, I think the introduction [that] it is characteristic for anelastic models that the pressure perturbation does not follow the**

**ideal gas law causes more confusion than clarification. I would omit it.**

The sentence has been changed to:
" Pressure perturbation $\pi$ is adjusted so that velocity field satisfies eq.(7). "

**P. 6, 13  14, Shima et al. (2009) were not the first to advocate the integration of the squared wet radius. See, e.g., J.-P. Chen (1992): Numerical simulations of the redistribution of atmospheric trace chemicals through cloud processes (Doctoral dissertation, Pennsylvania State University), especially his Eq. (3.81).**

We added a citation of the PhD thesis of J.-P. Chen.

**P. 6, ll. 18  19: In what sense is condensation a fast process here? I think you need to be more specific. Arnason and Brown (1971, doi:10.1175/1520-0469(1971)028¡0072:GOCDBC¿2.0.CO;2) showed that for condensation a timestep corresponding to the phase relaxation timescale is sufficient, i.e., about 1 s or even longer for clean clouds. The requirement for a 0.1 s timestep arises, in my eyes, from the rapid change in droplet radius during growth at small radii. This is a well-known feature of stiff differential equations, as it is the case for the diffusional growth equation for droplets. Furthermore, how do you know that a sub-stepping timestep of 0.1 s is sufficient? In similar simulations of Grabowski et al. (2011, doi:10.1016/j.atmosres.2010.10.020) an initial timestep of 10 -6 s that might increase to 0.1 s is used to integrate the diffusional growth equation (see their Appendix). Of course, they integrated the linear growth equation (dr/dt) and not the quadratic (dr 2 /dt) as done here. But additional stand-alone integrations of superdroplets with different aerosol masses and a prescribed supersaturation using different timestep lengths are necessary to verify if a 0.1 s sub-timestep is actually sufficient.**

What we mean by fast process is that it needs to be resolved on shorter time scales than other processes, precisely because small droplet radius changes rapidly by condensation. We state that the 0.1 s time step is sufficient based on tests we did in kinematic stratocumulus setup, in which concentration of cloud droplets converged for 0.1 s. It is possible that in other setups, e.g. with giant aerosols or stronger updrafts, a shorter time step would have to be used. The following text has been added to the manuscript:
" Condensation can rapidly change radii of small droplets. Therefore to correctly model condensation, in particular during the crucial moment of droplet activation, it is necessary to model condensation with a relatively short time step. Tests we performed in a kinematic 2D model of stratocumulus clouds have shown that number of activated droplets converges for condensation time step of around 0.1s. "

Based on our own experience and on personal communications with Shin-ichiro Shima, the relatively long time step of 0.1 s can be used thanks to the fact that we use the predictor-corrector algorithm described in the paper and that we solve growth equation for $r^2$ and not for $r$.

**P. 8, l. 8: Consider changing a pair to the same pair for clarity.**

Done

**P. 8, l. 26: RHS of what?**

For clarity, we changed that sentence to:
" In principle, liquid water is resolved by the SDM and could be diagnosed from the super-droplet size spectrum each time it is needed in the buoyancy term in eq. (3) or radiative term in eq. (4) "

**P. 8, ll. 30 31: These sentences contradict each other since the UWLCM contains an LCM and an LES. Therefore, specify [all] of the model dependent variables more precisely.**

Changed to:
" Eulerian dependent variables of the model are co-located. "

**P. 9, ll. 9 10, Fig. 2: Figure 2 confuses me. If only the shaded part is used as a coalescence cell, certain volumes filled with superdroplets are neglected in the collection process. However, I do not believe that this is what the authors are doing. Could it be the case that the lowest line of grid point always equals the first, and that the right-most column of grid point equals the left-most? In other words, how do the authors implement so-called ghost layers of grid points to facilitate a cyclic model domain?**

Superdroplets fill only the shaded region, as stated on p.9 l. 2:
" Super-droplets are restricted to the physical space, which is the shaded region in fig. 2. "
The domain is cyclic in horizontal directions, so left-most grid points (i.e. nodes of the primary grid) are equal to the right-most. This is not true for lower-most and upper-most grid points, because domain is rigid in the vertical direction. Ghost layers are implemented in such way that arrays stored in memory are larger than the grid shown in fig.2 and processes exchange values of ghost layers.

**P. 9, l. 18 19: Important for the formation of drizzle is the is the microphysical model, and usually not the LES dynamical core.**

In our understanding, a LES model of cloud needs to include some microphysical model. Therefore by LES model we mean dynamical core + microphysical model.

**P. 10, ll. 25 26: Please comment on these options if they are essential for the conducted simulations. If they are not essential, I would omit this sentence for clarity.**

Using other options would affect results, e.g. by giving more numerical diffusion. Therefore we chose to keep this sentence in case in future someone would try to reproduce our results.

**P. 10, ll. 30 32: State that turbulence in two dimensions behaves fundamentally different from turbulence in three dimensions.**

Added:
" However, it has to be kept in mind that the turbulence behavior in 2D is fundamentally different from 3D. "

**P. 10, l. 33 p. 11, l. 1: Small random perturbations are not the reason for the variability, it is a fundamental property of a chaotic system, reacting to small changes in the initial values.**

Because of the random perturbation, initial conditions are a little different for each run. Since the system is chaotic, small differences in initial conditions result in large differences after some time. Therefore we think that the statement that small initial perturbations cause variability is correct. If there were no random perturbations of initial conditions and microphysics were deterministic, each run would give the same result even though the system is chaotic, given that numerical calculations are exactly reproducible.

**Figs. 3 6: For the final version of this manuscript, please make sure that the location of the figures matches the text.**

We are doing our best. Additional formatting will need to be done afterwards, as the manuscripts in GMD are in a double column layout.

**P. 12, ll. 1 2: The entrainment is usually not calculated from the increase of the inversion height alone. Commonly, the subsidence velocity at cloud top height is subtracted.**

The sentence has been removed from the manuscript. Entrainment rate does take into account subsidence velocity. Definition of entrainment rate is now given in the caption of fig. 3 using symbols defined in Appendix A:

" Time series of the domain averaged liquid water path, entrainment rate (equal to $dz_i/dt + w_{LS}z_i$), maximum of vertical velocity variance, surface precipitation, concentration of cloud droplets in cloudy cells and cloud base height. "

**P. 12, ll. 23 24: I suggest rewriting this sentence to: [...] where the autoconversion efficiency increases with N SD .**

Done.

**P. 12, ll. 26 28: Since the characteristics of turbulence in two dimensions are fundamentally different from three dimensions, the better agreement with observations must be seen as purely coincidental.**

We agree. Still it is interesting to see.

**P. 13, l. 29: Why is the iLES approach responsible for the simulated behavior of the third moment of the vertical velocity?**

We suspected that based on Pressel et al. 2017 (doi:10.1002/2016MS000778). New simulations with SGS scheme, added in the revised paper, show that it is true - adding the SGS scheme gives skewness in agreement with reference models.

**P. 13, l. 33 p. 15, l. 2: Spurious cloud edge supersaturations are known to result in the artificial activation of cloud droplets at the top of stratocumulus (e.g., Stevens et al. 1996, doi:10.1175/ 1520- 0493(1996)124,1034:TSPOCE.2.0.CO;2). Physical activations are largely impossible there since the top of stratocumulus is not dominated by strong, long-lasting updrafts resulting in physical supersaturations.**

The sentence has been removed from the revised version.

**P. 17, ll. 1 2: Maybe it is worthwhile to add references to the models DHARMA and RAMS.**

Definitely, references have been added.

**P. 17, sec. 4.5: How is activation determined? I assume a droplet is considered activated when it exceeds a critical radius. This is a valid assumption if the aerosol is small, and diffusional growth is not kinetically limited. However, for aerosols smaller than 0.1 m, the typical timescale for activation is usually similar or even smaller than the timestep of the applied model, making the treatment of activation in UWLCM, DHARMA, and RAMS practically identical. The**

activation timescale becomes only important if the aerosol is large, typically larger than 0.1 m in radius, for which the critical radius exceeds a couple of micrometers. However, once located in a saturated environment, these inactivated particles exhibit behavior very similar to regularly activated droplets once their wet radius exceeds one micrometer, beyond which curvature and solute effects are usually negligible. Accordingly, the reduced susceptibility of aerosol activation on the cloud-base supersaturation maximum might also be just a result of the applied criterion for activation, which is not appropriate for the entire aerosol spectrum.

Droplet is considered activated when it becomes a cloud droplet, i.e. when its radius exceeds 0.5 $\mu$m (cloud droplet definition is in the caption of table A1). We also tested the definition assumed by the Referee, i.e. that activation happens when droplet radius exceeds the critical radius. Profiles of $N_c$ are very similar for both definitions.

In our opinion, UWLCM treats activation of small aerosol (smaller than 0.1 $\mu$m) differently than DHARMA or RAMS. Assume that after model timestep supersaturation in a given cell exceeds critical supersaturation. In DHARMA and RAMS that means that some droplets are activated. In UWLCM, condensation is resolved with a timestep of 0.1 s, shorter than timescale of activation of most aerosols. Therefore condensation can decrease supersaturation to values lower than critical supersaturation before any droplets exceed critical radius, hence it is possible that no droplets are activated.

**P. 17, ll.. 26 27: Please clarify: The cloud-base supersaturation maximum still causes activation in UWLCM, but it might not have an as immediate effect as in other cloud models because of the (presumably) applied criterion for activation (see last comment).**

See answer to the last comment.

**P. 19, ll. 10 11: I agree, that the number of superdroplets has no impact on domain-averaged quantities. However, it might be worthwhile to refer to the study of Schwenkel et al. (2018, doi:10.5194/gmd-11-3929-2018) in which small-scale effects of the superdroplets concentration are addressed. Technical Comments**

This issue is now addressed in section 4.2:
" For example, larger number of SDs would probably be needed in simulations in which SDs have more attributes, e.g. when modeling aqueous chemistry. Also, we expect that observables other than domain averages, e.g. related to the spatial structure of a cloud, are more sensitive to the number of SDs. Schwenkel et al. (2018) present in more detail how cloud structure depends on the number of SDs. "

**P. 5, l. 4: Change format of citation: [...] in Gillespie (1972), [...], not [...] in (Gillespie, 1972), [...]**

Done.

**P. 10, l. 16: Change format of citation: [...] in Ackerman et al. (2009), [...], not [...] in (Ackerman et al., 2009), [...]**
Done.

Answer to the Anonymous Referee #2.

**Major Comments**

**1. As the main components have been described elsewhere, and the coupling of the two parts seems to be rather straightforward, a stronger emphasis could be put on the model verification. You compare it to an ensemble of 11 reference models with quite some spread. But no one knows what the truth is. So far, I am not really sure what conclusions are to be drawn from your comparison and how I should interpret your results? Can you conclude anything e.g. from the fact that your model lies above or below the ensemble mean for some physical quantity? Better describe what you expect from your comparison exercise. Focusing on one specific test case gives only a snapshot of the models overall behavior and it is not clear how robust and general your findings are. It would be interesting to see how your model behaves in another well-chosen test case.**

The paper introduces a new model, which, like almost any other model, is based on the research of others. The equations solved, numerical methods, etc. have been used before. However, in order for other researchers to be able to use the model or to make a comparison with it, it is important to present these known components in one place. Besides providing such reference, new methods for coupling of the components are presented in the paper. We do not agree that these methods are straightforward. For example, Shima et al. (2009) uses other methods for spatial coupling and condensation substepping. Also, time stepping algorithm presented in Fig. 1 is not straightforward - time stepping could be done in many different ways.

The Dycoms RF02 setup is devised to reproduce observed clouds. Observations are the truth, albeit there are many difficulties in comparing modeling with observations. Nevertheless, models do reasonably well in reproducing observations. Therefore, if our model was far off from other models, that would indicate that something is wrong in it. Of course, if two models give slightly

different results it is impossible to say that one is better than the other. Comparing a model with Lagrangian microphysics with an ensemble of other models is also a novelty - we are not aware of similar studies.

Besides showing that the model gives results in general agreement with other models, 2D and 3D tests in the revised paper have additional purposes. 2D simulations are done to test sensitivity of the microphyscisc scheme - something of interest for other users of Lagrangian microphysics. 3D simulations are done to test sensitivity of the model to the description of SGS turbulence, including a SGS model for motion of Lagrangian computational particles.

**2. To be frank, resorting to the iLES approach comes in handy as you dont have to implement a SGS scheme. I could live with it if your model is purely Eulerian. As the Lagrangian model has no implicit numerical diffusivity (neither in spectral nor spatial space) and the iLES approach is not applicable in the microphysics part, SGS random perturbation velocities could be included in the transport equation of the superdroplets in order to mimic subgrid scale motions. However, without a proper SGS scheme that estimates TKE it is not straightforward to prescribe such perturbations. This shortcoming should be clearer mentioned.**

We agree that this has been a major drawback of the initial manuscript. Therefore the 3D simulations section now contains a comparison of ILES vs Smagorinsky vs Smagorinsky + SGS perturbation of superdroplets. For details, please see the answer to the Major Comment 1. of Referee #1.

**Minor comments**

**P1. Last row: Isnt libmpdata++ the dynamical core? What does it mean it is built on top of it?**

libmpdata++ is designed to be applicable to variety of problems. This means that some aspects, such as numerical integration procedure or details of the SGS scheme, have to be defined in the software that uses libmpdata++. In addition, all forcings are implemented in UWLCM. The sentence has been changed to:
"The dynamical core is implemented using the the libmpdata++ software library"

**p.4, l.22: Without defining what a collision between two SDs is, it makes no sense to say the probability needs to be increased. Please rephrase.**

We rephrased it from "collision" to "coalescence" of SDs. What a coalescence of SDs is is defined right after eq. (13) (eq. 17 in the revised manuscript) that presents how probability needs to be increased.

**p.5, l.14: I do not understand the inclusion of w LS. This would mean that the SDs move relative to the surrounding (Eulerian) air!?**

Large-scale subsidence is not included in air velocity, but is implemented as a RHS. SDs are advected with air velocity, i.e. without subsidence. Therefore the subsidence velocity needs to be separately added to the SD velocity.

**Sec 3.2.: The implementation of the various condensation algorithms is not clear to me. Given that new and old are known, you do a linear time interpolation between the two values. And the difference between the two approaches is the choice of the grid box from which you pick the values. What I stumble upon is the quantity new . Is it known beforehand? In my understanding, sub-stepping would simply mean that condensation (growth of droplets, depletion of water vapor and latent heat release) is treated with a smaller time step and clearly involves a dynamic update of the variables and q v in each sub- time step.**

As stated in section 3.2:
"$\psi_{\mathrm{new}}$ [are] values of Eulerian variables before the start of the substepping algorithm in the current time step".

Therefore it is known beforehand. $\psi_{\mathrm{new}} - \psi_{\mathrm{old}}$ is a change of Eulerian variables caused by sources other than condensation, e.g. surface fluxes, radiation, advection, etc. When substepping, we do a linear interpolation of this change and at each substep we add to that changes caused by condenation. An exact mathematical description is given in the Appendix B.

**Sec 3.3: I do not fully understand why you solve a prognostic equation for q l in the Eulerian model part. Wouldnt it suffice to diagnose q l from the SDs? I understand that q l is used for the computation of the buoyancy term (Eq. 6). Do you need it elsewhere? Can you estimate the error of using two different definitions of q l ? You write that you want to avoid an additional synchronization? Would this issue still matter in a parallelised implementation?**

$q_l$ is needed in buoyancy and radiation terms. Synchronization is needed precisely because our implementation is parallelised - calculations are done at the same time by CPU cores and by GPUs. As stated in Sec. 3.3, it would suffice to diagnose $q_l$ from SDs each time it is needed:
"In principle, liquid water is resolved by the SDM and could be diagnosed from the super-droplet size spectrum each time it is needed in the buoyancy term in eq. (3) or radiative term in eq. (4)."
However, the buoyancy term is integrated with a trapezoidal rule, hence we need to know liquid water at the next time step: $q_l(n+1)$. In principle we could wait for GPUs to finish calculating advection, subsidence and sedimentation

of SDs and then diagnose $q_l(n+1)$ from SDs and launch the pressure solver afterwards. However, plenty of computational time can be saved by running advection, subsidence and sedimentation in parallel with the pressure solver. This is achieved by adding the auxiliary $q_l$ field. To clarify our approach, the paragraph now reads:

" Liquid water is resolved by the SDM and $q_l$ could be diagnosed from the super-droplet size spectrum each time it is needed in the buoyancy term in eq. (3) or radiative term in eq. (4). Buoyancy is integrated with a trapezoidal scheme, which requires $q_l$ after advection to be known. In a straightforward implementation, in which $q_l$ is diagnosed from SDs after advection of SDs, pressure solver calculations can only be started after advection of SDs has been calculated. Then, there is little parallelism of calculations on GPUs and CPUs. To achieve more parallelism, we introduce an auxiliary Eulerian field for $q_l$. Value of $q_l$ is diagnosed from SDs once per timestep, after condensation calculation. Then, $q_l$ advection is done using a first-order accurate upwind scheme. Using the auxilliary $q_l$ field, it is possible to calculate coalescence and motion of SDs simultaneously with calculations of advection of Eulerian fields and of the pressure problem. "

We expect the error associated with this procedure to be low, because $q_l$ is diagnosed from SDs at each time step.

**Sec 4.1: Can you comment why you use a split definition (Hall +Davies) for the collision efficiencies?**

Hall (1980) does not give collision efficiencies for collisions of droplets that are both smaller than 10 $\mu$m, therefore for such collisions we use values from Davies (1972).

**Sec 4.2: In particular, the differences between the per-cell and per-particle approach are so small that I am not fully convinced that the one is superior over the other one. It would also help to see the spread of the 10-member ensemble of a specific 2D simulation. Is it really significant that in the one case the N c -profiles slightly decrease with altitude, whereas in the other case they slightly increase? Can you be sure that in other test case, your finding (superiority of the per-particle) would be the same? This is one example why I recommend a second test case.**

In the revised manuscript, substepping algorithms are tested using kinematic approach, i.e. both simulations are run with the same flow field. Results of single runs are compared, without averaging over an ensemble. This improved test case has shown that the *per-cell* algorithm works a little better for stratocumulus clouds, contrary to what we initially concluded. We also added a second test case for substepping that represents idealized advection of cloud edge. In that case, the *per-particle* algorithm works much better. These tests are described in Appendix B of the revised manuscript.

**Typos, language issues and other formal things:**

**In general, the usage of articles a and the is not correct on several occasions. Sometimes you miss the article, sometimes it is misplaced. Please try your best, the rest will be handled by Copernicus services.**

We are doing our best.

**The Exner function pi should be defined close to Eq. 3**

$\pi$ is pressure perturbation, what is now stated in the sentence following Eq. 3. Definition of it remains in the table in the Appendix A.

**p.4, l.16: collisionS**

Fixed.

**There is a difference between which and that: https://www.wisegeek.com/what-is-the- difference-between-that-and-which.htm Accordingly, which in p.4, l.19 and l.24 must be replaced by that. There might be more such mistakes.**

Thank you for this language tip. Several more occurrences of "which" have been replaced with "that".

**p.4, l. 23: dropletS**

Fixed.

**p.12, l.10. not sure if VAR is self-explaining?**

It is now defined in the caption of Fig. 3.

**p.12, l.31: visible IN**

Fixed.

**p.13, l.11: impact IN 3D simulations than IN 2D simulations**

Fixed.

**Caption of Fig. 4: Please correct On the vertical axis is height ...**

Changed to:
" Vertical axis is altitude normalized by inversion height."

**P.17, l.17: concentration**

Fixed.

**References**

Schwenkel, J., Hoffmann, F., and Raasch, S.: Improving collisional growth in Lagrangian cloud models: development and verification of a new splitting algorithm, Geoscientific Model Development, 11, 3929–3944, 2018.

Shima, S.-i., Kusano, K., Kawano, A., Sugiyama, T., and Kawahara, S.: The super-droplet method for the numerical simulation of clouds and precipitation: A particle-based and probabilistic microphysics model coupled with a non-hydrostatic model, Quarterly Journal of the Royal Meteorological Society, 135, 1307–1320, 2009.

Smolarkiewicz, P. K., Khnlein, C., and Wedi, N. P.: A consistent framework for discrete integrations of soundproof and compressible PDEs of atmospheric dynamics, Journal of Computational Physics, 263, 185 – 205, https://doi.org/https://doi.org/10.1016/j.jcp.2014.01.031, URL http://www.sciencedirect.com/science/article/pii/S0021999114000588, 2014.

---

## Referee Report (RR1)

**2$^{nd}$ Review of "University of Warsaw Lagrangian Cloud Model (UWLCM) 1.0: a modern Large-Eddy Simulation tool for warm cloud modeling with Lagrangian microphysics" by Dziekan et al. (doi:10.5194/gmd-2018-281)**

The revised manuscript addresses most of my comments thoroughly. However, I have some minor comments before the manuscript can be accepted for publication.

Note that I will reference the lines in the tracked-changes version of the manuscript.

**Minor Comments**
P. 1, ll. 9 – 11: I agree that a time step of the order of 0.1 s is necessary for condensation and collection. However, I would clearly state that this time step can be facilitated by substepping, and not by running the entire model using a time step of 0.1 s.

P. 3, l. 25: To avoid confusion, state clearly what is contained in each dimension of $F_u$. In particular, state that large-scale subsidence is not applied to the vertical component of $F_u$.

P. 5, ll. 23 ff.: I assume that $K_{i,k}$ is the collection and not the coalescence kernel. Collection is the product of collision and subsequent coalescence. Both processes need to happen that two droplets merge into one, i.e., droplets need to collide (which is not always happening due to hydrodynamic interactions of the droplets) and to coalesce (which is not always happening due to surface tension effects). That is also the reason why $K_{j,k}$ usually contains two efficiencies, one for the collision of (super-) droplets j and k and one for the coalescence of (super-) droplets j and k. I recommend writing about collection (or collision-coalescence) if the authors address the eventual merging of the droplets (http://glossary.ametsoc.org/wiki/Collection_efficiency).

P. 6, l. 23: For clarity, write about the "transport by the mean air flow" and not "advection by air flow". Advection usually refers to the transport in an Eulerian framework, which is not the case here.

P. 7, ll. 25 – 28: Of course, condensation changes the radius of small droplets rapidly. However, this has only a minuscule effect on the overall supersaturation, since although the small droplets change their radius rapidly they do not deplete a large amount of supersaturation which is largely produced by dynamics. Therefore, if the interplay of the depletion of supersaturation by the integral droplet growth and the production of supersaturation by dynamics is not represented correctly, activation will be overestimated. However, substepping is not the only approach to solve this problem. In fact, it has been solved more than 40 years ago: Clark (1973, doi: *10.1175/1520-0469(1973)030<0857:NMOTDA>2.0.CO;2*) developed an approach to analytically solve for the mean supersaturation during a time step that can be used to model activation successfully. This approach has been used in many other detailed cloud models before.

P. 9, ll. 27 – 34: Unterstrasser et al. (2017, doi: 10.5194/gmd-10-1521-2017) showed successfully that longer time steps can be used for collection. However, I do not doubt the author's results. But I assume the reason for the need for substepping is the unusually small number of superdroplets used in their simulations, which overestimates the growth of individual superdroplets, which can be mitigated by substepping. This should be stated as such. (The authors use 40 superdroplets per grid box, most other studies use about 100.)

Fig. 2: To avoid confusion, label the axes in the figure.

P. 19, l. 25: Isn't there a year to the RAMS technical description?

P. 21, ll. 14 – 18: I agree that the width of the droplet spectra in the Lagrangian cloud model might be too narrow since they miss certain processes that are implicitly "included" by numerical diffusion in Eulerian bin models. However, naming specific processes can be misleading. Moreover, the correct

consideration of "lucky droplets" might only be possible in Lagrangian cloud models, and can never be captured in Eulerian bin models that are based on the Smoluchowski equation that does not include statistical effects.

P. 21, ll. 23- 26: I recommend to cite Hoffmann (2016, doi:*10.1175/MWR-D-15-0234.1*). His study on spurious supersaturations in Lagrangian cloud models stated the explanations given here.

P. 22, l. 20: I think it is worthwhile to note that the timestep of 0.1 s can be reached by substepping, not by running the entire model with a timestep of 0.1 s.

---

## Author Response (AR2)

**Response to the 2$^{\text{nd}}$ Review of "University of Warsaw Lagrangian Cloud Model (UWLCM) 1.0: a modern Large-Eddy Simulation tool for warm cloud modeling with Lagrangian microphysics" by Dziekan et al. (doi:10.5194/gmd-2018-281)**

June 3, 2019

We are grateful to the reviewer for his suggestions.

**Minor Comments**

**P. 1, ll. 9 11: I agree that a time step of the order of 0.1 s is necessary for condensation and collection. However, I would clearly state that this time step can be facilitated by substepping, and not by running the entire model using a time step of 0.1 s.**

The following has been added:
"Such short time steps are achieved by substepping, as the model time step is typically around 1 s."

**P. 3, l. 25: To avoid confusion, state clearly what is contained in each dimension of F u . In particular, state that large-scale subsidence is not applied to the vertical component of F u .**

We feel that the statement "The terms $F_*$ represent a total forcing due to surface fluxes, radiative heating/cooling, large-scale subsidence and absorbers" is already clear, because large-scale subsidence *is* applied to all variables, including vertical velocity.

Consider a situtaion, in which the large-scale subsidence velocity $w_{LS}$ is directly added to the velocity field: $\vec{u_t} = \vec{u} + (0, 0, w_{LS})$, where $\vec{u}$ is the resolved velocity field. Then, assuming no additional sources, conservation equation for a scalar $\psi$ is:

$$\partial_t \psi + \vec{u_t} \cdot \nabla \psi = 0, \tag{1}$$

what is equal to:
$$\partial_t \psi + \vec{u} \cdot \nabla \psi = -w_{LS} \partial_z \psi. \tag{2}$$

Ackerman et al. (2009) define that large-scale subsidence should be included as a right-hand side, i.e. the second equation should be used. Then, all Eulerian variables are affected by large-scale subsidence, but large-scale subsidence velocity is not added to the resolved velocity field. Therefore, super-droplets need to by advected by the resolved velocity plus $w_{LS}$.

**P. 5, ll. 23 ff.: I assume that K i,k is the collection and not the coalescence kernel. Collection is the product of collision and subsequent coalescence. Both processes need to happen that two droplets merge into one, i.e., droplets need to collide (which is not always happening due to hydrodynamic interactions of the droplets) and to coalesce (which is not always happening due to surface tension effects). That is also the reason why K j,k usually contains two efficiencies, one for the collision of (super-) droplets j and k and one for the coalescence of (super-) droplets j and k. I recommend writing about collection (or collision-coalescence) if the authors address the eventual merging of the droplets (http://glossary.ametsoc.org/wiki/Collection_efficiency).**

To make it clear, we replaced "coalescence" with "collision-coalescence".

**P. 6, l. 23: For clarity, write about the transport by the mean air flow and not advection by air flow. Advection usually refers to the transport in an Eulerian framework, which is not the case here.**

We have changed this sentence accordingly.

**P. 7, ll. 25 28: Of course, condensation changes the radius of small droplets rapidly. However, this has only a minuscule effect on the overall supersaturation, since although the small droplets change their radius rapidly they do not deplete a large amount of supersaturation which is largely produced by dynamics. Therefore, if the interplay of the depletion of supersaturation by the integral droplet growth and the production of supersaturation by dynamics is not represented correctly, activation will be overestimated. However, substepping is not the only approach to solve this problem. In fact, it has been solved more than 40 years ago: Clark (1973, doi: 10.1175/1520-0469(1973)030¡0857:NMOTDA¿2.0.CO;2) developed an approach to analytically solve for the mean supersaturation during a time step that can be used to model activation successfully. This approach has been used in many other detailed cloud models before.**

We acknowledge that other, more efficient methods than substepping, can give reasonably good results. However, these methods make assumptions or use

approximations that are not needed in substepping. Our goal is to have a LES model with as little approximations in microphysics as possible, hence we use substepping. One advantage of having a precise model is that it can be used to study errors associated with using more approximate methods. A good example is the maximum in cloud droplets concentration near cloud base produced by bin microphysics, that is not seen in our model. Our interpretation is that the maximum is caused by approximate methods for modeling activation used in bin microphysics.

In section 5d. of Clark (1973) it is stated that the method with time-averaged value of supersaturation "cannot be applied, for instance, when intense nucleation is occurring, ...". In that case, the explicit method described in 5c., that includes substepping, should be used. Also, treatment of nucleation in Clark (1973) can lead to negative concentrations of cloud droplets (section 5e). Then, an additional numerical trick called "hole-filling" is used. Clark 1973 justifies using "hole-filling" and time-averaged supersaturation by comparing results of these approximate methods with results of explicit modeling in idealized setups. Today, more than 40 years later, it is possible to use explicit modeling in large-scale simulations.

**P. 9, ll. 27  34: Unterstrasser et al. (2017, doi: 10.5194/gmd-10-1521-2017) showed successfully that longer time steps can be used for collection. However, I do not doubt the authors results. But I assume the reason for the need for substepping is the unusually small number of superdroplets used in their simulations, which overestimates the growth of individual superdroplets, which can be mitigated by substepping. This should be stated as such. (The authors use 40 superdroplets per grid box, most other studies use about 100.)**

Untersrasser et al. (2017) showed that longer time steps can be used for collection in box model simulations of pure collision-coalescence. We also tested our algorithm in a box model and came to the same conclusion as Unterstrasser et al. : that averages over large ensembles agree with the Smoluchowski equation even for time steps of the order of couple of seconds and few superdroplets. Therefore we were surprised to find that in LES the time step needs to be shorter. This subject was discussed during the recent Cloud Modeling Workshop (`http://ww2.ii.uj.edu.pl/~arabas/workshop_2019/`). Other groups also found that in LES the collection time step has to be shorter than in a box model.

Increasing the number of superdroplets might allow us to use longer collection time steps, but definately more than 100 super-droplets would need to be used. Our 2D simulations (Figure 3) show that increasing the number of super-dropletes to 1000 does not help the issue - collection still needs to be resolved with a 0.1 s time step.

**Fig. 2: To avoid confusion, label the axes in the figure.**

Done.

**P. 19, l. 25: Isnt there a year to the RAMS technical description?**

No, there is no date to it. The technical manual probably changes with changing versions of RAMS. We cite it because it is the best description of RAMS that we could find.

Details about DHARMA and RAMS simulations presented in Ackerman et al. (2009) are given in Appendix B to that paper. We added a sentenc refering the readers to it:

"More details about DHARMA and RAMS simulations of the DYCOMS RF02 case can be found in the Appendix B of Ackerman et al. (2009)."

**P. 21, ll. 14 18: I agree that the width of the droplet spectra in the Lagrangian cloud model might be too narrow since they miss certain processes that are implicitly included by numerical diffusion in Eulerian bin models. However, naming specific processes can be misleading. Moreover, the correct consideration of lucky droplets might only be possible in Lagrangian cloud models, and can never be captured in Eulerian bin models that are based on the Smoluchowski equation that does not include statistical effects.**

To make it clear that we only name some examples of such processes, and not give a comprehensive list, the sentence now reads:

"Possibly, precipitation is affected by some physical processes that are not currently modeled by UWLCM. The list of such unresolved processes that could affect precipitation includes, but is not limited to, the following: SGS turbulence affecting condensation and coalescence of droplets, lucky droplets effect and giant CCN initiating rain formation."

We agree that using Smoluchowski equation the "lucky droplets" effect can not be modeled. In the paper we do not argue that bin models do include this effect. We only stress that our Lagrangian model does not include it either.

**P. 21, ll. 23- 26: I recommend to cite Hoffmann (2016, doi:10.1175/MWR-D-15-0234.1). His study on spurious supersaturations in Lagrangian cloud models stated the explanations given here.**

The citations has been added.

**P. 22, l. 20: I think it is worthwhile to note that the timestep of 0.1 s can be reached by substepping, not by running the entire model with a timestep of 0.1 s.**

[revised manuscript text omitted]